# Data subsampling for Poisson regression with $p$th-root-link

**Han Cheng Lie**
Institut für Mathematik
Universität Potsdam
Germany
hanlie@uni-potsdam.de

**Alexander Munteanu**
Department of Statistics
TU Dortmund University
Germany
alexander.munteanu@tu-dortmund.de

## Abstract

We develop and analyze data subsampling techniques for Poisson regression, the standard model for count data $y \in \mathbb{N}$. In particular, we consider the Poisson generalized linear model with ID- and square root-link functions. We consider the method of *coresets*, which are small weighted subsets that approximate the loss function of Poisson regression up to a factor of $1 \pm \varepsilon$. We show $\Omega(n)$ lower bounds against coresets for Poisson regression that continue to hold against arbitrary data reduction techniques up to logarithmic factors. By introducing a novel complexity parameter and a domain shifting approach, we show that sublinear coresets with $1 \pm \varepsilon$ approximation guarantee exist when the complexity parameter is small. In particular, the dependence on the number of input points can be reduced to polylogarithmic. We show that the dependence on other input parameters can also be bounded sublinearly, though not always logarithmically. In particular, we show that the square root-link admits an $O(\log(y_{\max}))$ dependence, where $y_{\max}$ denotes the largest count presented in the data, while the ID-link requires a $\Theta(\sqrt{y_{\max}/\log(y_{\max})})$ dependence. As an auxiliary result for proving the tightness of the bound with respect to $y_{\max}$ in the case of the ID-link, we show an improved bound on the principal branch of the Lambert $W_0$ function, which may be of independent interest. We further show the limitations of our analysis when $p$th degree root-link functions for $p \geq 3$ are considered, which indicate that other analytical or computational methods would be required if such a generalization is even possible.

## 1 Introduction

Random sampling is arguably one of the most popular approaches to reduce large amounts of data to save memory, runtime, and further downstream resources such as communication bandwidth and energy. In contrast, classic statistical learning theory often uses uniform sampling and provides only asymptotic approximation guarantees. These guarantees often require strict assumptions such as i.i.d. data and for model assumptions to be met exactly. However, data collected from real applications often violate these conditions: only finite samples are available, independence might not be satisfied, and the model may deviate from reality. When model fitting algorithms are applied to such data, we are not only interested in reducing the above-mentioned resource requirements, but also in providing rigorous worst-case guarantees on approximation.

Arguably, the most popular approach is the Sensitivity Framework [26, 19], which provides a general-purpose importance sampling scheme that yields a weighted subsample — or *coreset* — that given a data matrix $X$ approximates some loss function $f(X\beta)$ within a factor $(1 \pm \varepsilon)$ for any query point $\beta$. This guarantee can be stated as follows: a significantly smaller subset $K \subseteq X, k := |K| \ll |X|$

together with corresponding weights $w \in \mathbb{R}^k$ is a $(1 \pm \varepsilon)$-coreset for $X$ if it satisfies

$$\forall \beta \in \mathbb{R}^d \colon |f(X\beta) - f_w(K\beta)| \leq \varepsilon \cdot f(X\beta). \tag{1}$$

We point the interested reader to [37, 35, 32] for a gentle introduction and overview on coresets.

Our aim is of course not only to obtain good approximation accuracy as stated in Equation (1), but also for the subsample achieving the bound to be sublinear in the input size. Unfortunately, most generalized linear models do not admit strongly sublinear data summaries with reasonable approximation guarantees [36, 31]. This also holds for the Poisson models considered in this paper.

To go beyond the worst-case setting and enable meaningful data reduction, a natural approach is to parameterize the analysis with a quantity that captures the fit of data to the statistical model and quantifies the achievable size of succinct data summaries [36]. Another ingredient that is commonly used to tackle data reduction for generalized linear models is to relate their loss to $\ell_p$ norms, for which $\ell_p$ sensitivities or leverage scores yield viable importance sampling distributions [16, 12, 36, 34].

## 1.1 Our contributions

We provide the first rigorous analysis for $(1 \pm \varepsilon)$-approximate data reduction for Poisson models:

1. We show $\Omega(n)$ lower bounds against coresets for Poisson regression (Lemma 6.1), showing that changing the link function alone does not resolve the problem of bounding the complexity of coresets for the log-link, which is incompressible. Our lower bound extends to arbitrary data reduction techniques up to a $\log(n)$ factor (Lemma 6.2).

2. We introduce a novel complexity parameter $\rho$ that captures the compressibility of data under a Poisson $p$th-root-link model (Equation (2)). This parameter corresponds naturally to the statistical model assumptions and establishes a relationship between these assumptions and an optimization perspective of the compressibility problem.

3. We conduct a parameterized analysis, showing that sublinear coresets exist under the statistically natural assumption of small $\rho$ parameter (Theorem 3.8), and using a novel domain shift idea for their optimization (Theorem 4.2).

4. We prove a square root upper bound for the Lambert $W_0$ function over $[-1/e, 0)$ (Lemma 6.3) that allows us to prove tight bounds for the slope of a linear lower envelope (Lemma 6.4). This justifies an $\tilde{\Theta}(\sqrt{y_{\max}})^1$ dependence for the ID-link, which is contrasted by an $O(\log(y_{\max}))$ dependence for the square root-link.

5. We show the limitation of our domain shifting approach, showing that the error of this method cannot be bounded to give the required $(1 + \varepsilon)$ approximation for $p \geq 3$ (Lemma 6.5). This indicates a limitation to the common choices $p \in \{1, 2\}$, and suggests that different techniques than the ones we develop below may be needed to overcome this limitation.

## 1.2 Our techniques

The general outline of our analysis follows the established method of sensitivity sampling [26, 19]. Several steps along this outline require novel ideas due to peculiarities of the Poisson loss function defined in Equations (3) and (4) below. A VC dimension bound of $d^2$ is easy to obtain by counting the number of arithmetic operations required to compare an individual loss to a given threshold as a measure of complexity [4]. A near-linear $\tilde{O}(d/\varepsilon)$ bound is obtained by a more fine-grained analysis, by grouping and rounding the associated count data (respectively, sensitivity scores) to powers of $(1 + \varepsilon)$ (resp. to powers of 2). This results in a surrogate loss function that admits group-wise linear VC dimension, by splitting the domain of each loss function at its unique global minimum into two regions such that the restriction of the function to each region is monotone, and connecting the resulting construction to hyperplane classifiers. Approximating the surrogate finally implies the desired $(1 \pm \varepsilon)$ approximation for the original loss as well.

For bounding the sensitivities, the domain of the loss function $g(z)$ is split into three intervals: 1) one interval consisting of 'moderate' values of $z$, such that the $g(z)$ values are bounded above and below within constants and can be treated using simple uniform sampling; 2) one interval where $z$ has large

---

[1] $\tilde{O}(\cdot)$ hides lower order log terms, analogous notation applies to all Landau symbols.

values, in which case $g(z)$ is closely related to $z^p$; 3) one interval, where $z$ is close to 0, in which case the negative logarithm dominates the loss.

Tackling the interval in 2) requires relating the loss function $g_y(z)$ to $z^p$. Specifically we would like to bound $z^p \geq g_y(z) \geq z^p/\lambda$ for sufficiently large $z$ and for some value of $\lambda$. This requires special care, since the loss function $g_y(z)$ is translated polynomially towards larger values of $z$ with growing $y$, but the minimum of $g_y(z)$ grows only logarithmically with $y$. Informally speaking, the loss function 'widens', and its minimum 'moves mainly to the right', so for large $y$, we would need a very flat lower bound, which requires large $\lambda \approx y$ (ignoring polylogarithmic terms). However, this is undesirable, since $\lambda$ appears to be a crucial parameter for bounding the subsample size. Specifically, this would yield a $\sum_{i \in [n]} y_i = \Omega(n)$ dependence in the coreset size. So instead, we bound the loss roughly as $z^p \geq g_y(z) \geq (z - y^{1/p})^p/\lambda$, which amounts to translating the lower envelope with growing $y$ as well. Additionally, we introduce a novel complexity parameter $\rho$ which balances the translated $(z - y^{1/p})^p$ lower bound with the $z^p$ upper bound. We stress that these translation and balancing arguments are not artificial or just used to make calculations go through, but they are naturally consistent with the statistical model (see the discussion below Equation (2)). This proof strategy eventually captures the loss function within interval 2) more closely and yields sublinear bounds for $\lambda$ as well.

We remark that in contrast to $\mu$-complexity in previous work [36, 34], a bounded balancing complexity parameter $\rho$ does *not* handle the asymmetry between intervals 2) and 3). Tackling the interval in 3) thus also requires completely new ideas, as the negative logarithm has an infinite asymptote at 0, which we exploit to prove $\tilde{\Omega}(n)$ lower bounds on subsample size in Lemma 6.1 and Lemma 6.2. Such asymptotes have not been mentioned or analyzed in previous related work on sensitivity sampling for GLMs. To circumvent the lower bound, we avoid this interval by introducing a novel *domain shifting* approach, requiring all feasible solutions to satisfy $z > \eta$ for a suitable $\eta > 0$ for optimization. Choosing $\eta$ in the order of $\varepsilon$, we can argue that the solution in the shifted domain is a $(1 + \varepsilon)$ approximation. Avoiding the asymptote enables a coreset construction for the shifted domain.

We believe that the domain shifting approach is necessary: if an instance consists of the extreme points on the convex hull, and all but a small (sublinear) number of points are separated by an $\varepsilon$ distance to the boundary, then required structure is already in the data. But if non-extremal points are allowed to be arbitrarily close to the boundary, and we do not shift the domain, then we will not avoid high sensitivity points that are strictly inside the convex hull. Then the coreset size would necessarily depend on the distance of these non-extremal points to the boundary, and crucially on the number of points that are very close to the boundary of the convex hull, which again can be $\Omega(n)$.

Exploiting the asymmetry between the intervals 2) and 3) where the loss exhibits $z^p$ and $-\log(z)$ growth respectively, we prove $\tilde{\Omega}(n)$ lower bounds on subsample size, by adapting known reduction techniques [31, 36, 41]. We also provide lower bounds on parameters used in our analysis, showing their tightness. In particular, the aforementioned $\lambda$ slope parameter is of size $\Theta(\sqrt{y/\log(y)})$ in the case $p = 1$. The proof is conducted by an exact characterization of the tangent point between our linear lower envelope and the loss function. Since this requires balancing between $z$ and $\log(z)$ and examining the $\log(y!)$ function, further bounds rely on Stirling's approximation and the principal branch of the Lambert $W_0$ function. Recent bounds in [38] for the Lambert $W_0$ function imply our square root upper bound but only a cubic root lower bound. We thus significantly tighten their bound in an appropriate interval. This result may be of independent interest, since the Lambert function cannot be expressed in terms of elementary functions and has many important applications in various fields. As a result, we obtain a matching square root lower bound on $\lambda$ in our context.

## 1.3 Related work

Classic work on data subsampling started with linear $\ell_2$ regression [16] and was extended to linear $\ell_p$ regression [12]. More recently, the study continued with non-linear transformations such as in generalized linear regression models. The first guaranteed finite subsample bounds for logistic regression appeared in [36], while impossibility results for Poisson regression were given in [31]. Research on generalized linear models was continued for Probit regression [34]. Asymptotic properties of subsampling for generalized linear models, including for Poisson regression, were studied in [3, 27]. A finite-sample size result is given in [3, Theorem 5] that exhibits $O(\sqrt{d})$ approximation error. [13]

studied a sampling-based feature space reduction for a wide array of generalized linear models with additive errors. However, parts of their assumptions specifically do not apply to Poisson regression.

## 2 Preliminaries and the Poisson $p$th-root-link model

Poisson regression models aim to predict a count variable $Y \in \mathbb{N}_0$ using a generalized linear model with link function $h : \mathbb{R} \to \mathbb{R}$, i.e.,
$$h(\mathbb{E}(Y \mid x)) = x\beta,$$
where $x = (1, x^{(1)}, \ldots, x^{(d-1)}) \in \mathbb{R}^d$ is a *row* vector, and $\beta \in \mathbb{R}^d$ is a column vector carrying the model parameters that in particular include an intercept $\beta_1$ [30]. Common choices for $h$ are the canonical log-link $h(v) = \ln(v)$, the ID-link $h(v) = v$, and the root-link $h(v) = v^{1/2}$. The latter two can be cast into a general framework by introducing the $p$th-root-link $h(v) = v^{1/p}$, for any $\mathbb{R} \ni p \geq 1$, where the ID-link and root-link correspond to $p \in \{1, 2\}$ [10].

Subsampling for the log-link is not possible with the multiplicative $(1 + \varepsilon)$ error guarantees that we aim for, since it entails preserving the $\exp(x\beta)$ function [31]. We will also show impossibility results for the $p$th-root-link. However, we parameterize our analysis with a data-dependent parameter that reflects naturally how well the realized data distribution is captured by the Poisson model. This parameter is inspired from previous work [36]: we will refer to data $X, y$ as being '$\rho$-complex', if there exists a $0 < \rho < \infty$ such that denoting by $x_j$ the $j$-th row of the data matrix $X \in \mathbb{R}^{n \times d}$ and by $y_j$ the $j$-th entry of the vector $y \in \mathbb{N}_0^n$, it holds that

$$\sup_{\beta \in \mathbb{R}^d} \frac{\sum_{j=1}^n |x_j\beta|^p}{\sum_{j=1}^n |x_j\beta - y_j^{1/p}|^p} \leq \rho. \tag{2}$$

We may interpret the parameter $\rho$ as follows. The rate parameter of the predicted Poisson distribution is $\mathbb{E}(Y \mid x) = (x\beta)^p$. Hence, the mean and variance of $Y_j$ given $x_j$ is $(x_j\beta)^p$, for each $j = 1, \ldots, n$. To obtain the maximum likelihood estimator of $\beta$, we seek $\beta$ such that for every $j = 1, \ldots, n$, $x_j\beta$ is as close as possible to $y_j^{1/p}$, since $y_j^{1/p}$ minimizes the $j$th summand $g_{y_j}$ of the loss function specified in Equations (3) and (4) below. However, choosing $\beta$ so that $|x_j\beta - y_j^{1/p}|$ is small will imply that each summand in the numerator of Equation (2) will be close to $y_j^{1/p}$. In this case, the variance of the $(Y_j)_{j=1}^n$ will not be captured effectively by the Poisson model, and $\rho$ will be large. Thus, smaller values of $\rho$, i.e., values of $\rho$ that are closer to 1, indicate that the true data distribution is better captured by the Poisson model. Thus the $\rho$ parameter in Equation (2) plays a similar role of quantifying model fit as the $\mu_w(X)$ parameter from [36, Section 2]; see in particular the comments at the end of that section.

Assuming that the value of $\rho$ is small allows us to use the proximity of the negative log-likelihood to $\ell_p$ norms, together with some novel optimization ideas involving a shifted domain. This yields the first provable finite and sublinear subsample size with rigorous $(1 + \varepsilon)$ approximation guarantee. We focus on the special cases $p \in \{1, 2\}$ since they are the most popular (in fact the only practical) alternatives to the intractable log-link [10, 31]. The ID-link has been used in epidemiology [40, 29]. The root-link function has been applied to forecasting for queueing systems [39] and to account for misspecification bias in maximum likelihood estimation [17]. When the estimated mean count of the data is zero, then the canonical log-link causes problems that can be avoided by using the root-link; see e.g. [28, Section 5.4]. We also discuss in our lower bounds section other choices for $p$, and show that for any natural number $p \geq 3$ the bound implied by our novel shifting idea must fail. This bound is crucial to obtain our final approximation, indicating that other methods would be required to tackle a generalization for $p \geq 3$, if this is even possible.

Given parameters $\beta$ and an input $x$, the rate parameter of the predicted Poisson distribution is
$$\mu := \mathbb{E}(Y \mid x) = (x\beta)^p,$$
which corresponds to its mean and variance. Its probability mass function is $\mathbb{P}(Y = y) = \text{Poisson}(y \mid x\beta) = \frac{\mu^y e^{-\mu}}{y!} = \frac{(x\beta)^{py} e^{-(x\beta)^p}}{y!}$. Given a set of i.i.d. observations expressed as the rows $x_i$ of a data matrix $X \in \mathbb{R}^{n \times d}$ with corresponding labels $y \in \mathbb{N}_0^n$ we can obtain a maximum likelihood estimate of the parameter $\beta$ by minimizing the negative log-likelihood, which takes the form

$$f_y(X\beta) := \sum_{i=1}^n g_{y_i}(x_i\beta) = \sum_{i=1}^n (x_i\beta)^p - py_i \ln(x_i\beta) + \ln(y_i!) \tag{3}$$

where

$$g_{y_i}(x_i\beta) := (x_i\beta)^p - py_i \log(x_i\beta) + \log(y_i!). \tag{4}$$

For any $p$th-root-link, the loss function includes a $\log(x\beta)$ term, which restricts the feasible set to all $\beta$ such that for all $x_i, i \in [n]$ it holds that $x_i\beta > 0$. We note that for summands corresponding to $y_i = 0$, the function $g_0(z)$ simplifies to $g_0(z) = z^p > 0$ with well-known properties of the $\ell_p$ norm. We thus focus on summands $g_{y_i}$ for $y_i \in \mathbb{N}$ below.

For arbitrary $y \in \mathbb{N}$, the function $g_y(z) = z^p - py \ln(z) + \ln(y)$ on $\mathbb{R}_{>0}$ is strictly convex with first and second derivatives $g_y'(z) = pz^{p-1} - \frac{py}{z}$ and $g_y''(z) = p(p-1)z^{p-2} + \frac{py}{z^2} > 0$ respectively. Thus $g_y(z)$ decreases on the interval $z \in (0, y^{1/p})$, increases on $z \in (y^{1/p}, \infty)$, and has a unique minimizer at $z^* = y^{1/p}$ with corresponding value $y - y\log(y) + \log(y!) \approx \frac{1}{2}\log(y) + \Theta(1) \geq 1$ by Stirling's approximation. We shall use the following lower bounds to capture the $y$-dependence.

**Lemma 2.1.** *It holds for all $z \in \mathbb{R}_{>0}, p \in [1,\infty), y \in \mathbb{N}$ that*

$$g_y(z) = z^p - py\log(z) + \log(y!) \geq \max\left\{1, \frac{1}{3}(1 + p\log(z))\right\}.$$

The next two results bound the individual loss contributions from above and below by roughly a value of $z^p$. For the lower bound, however, we note that as the value of $y$ grows, the loss function is translated polynomially towards larger $z$ values, since its minimum is attained at $z^* = y^{1/p}$. However, by Lemma 2.1, and the properties above, the increase of $g_y(z^*)$ is only logarithmic in $z^*$, and thus also logarithmic in $y$. Denote by $\lambda$ the scaling parameter of the lower bound on $g_y$ given in Lemma 2.2. Then the logarithmic growth of $g_y(z^*)$ implies that we would need $\lambda \approx y/\log(y)$. Unfortunately, the value of $\lambda$ will affect the coreset size, which is undesirable, since it can become linear simply due to large values of $y$. We thus require a sublinear dependence on $y_{\max}$, i.e., the largest value of $y$ presented in the data. To this end, we shift the lower envelope by the minimizer $z^* = y^{1/p}$. The value of $\lambda$ can subsequently be bounded in a desirable way, but differs significantly depending on the value of $p$: in the case $p = 1$ we prove $\lambda \in O(\sqrt{y/\log(y)})$ to be sufficient, while the case $p = 2$ even constant $\lambda = 1$ will suffice. In Section 6, we will show a separation by a superconstant and matching square root lower bound on the value of $\lambda$ in the dominating case $p = 1$.

**Lemma 2.2.** *For any $p \geq 1$ and $y \in \mathbb{N}$ it holds that $z^p \geq g_y(z)$ for $z > y^{1/p}$. If $p = 1$, then for some $\lambda \in O(\sqrt{y/\log(y)})$, it holds that $g_y(z) \geq \frac{(z-y^{1/p})^p}{\lambda}$ for $z > y^{1/p}$.*

**Lemma 2.3.** *Let $p \geq 2$ and $y \in \mathbb{N}$, and $\lambda = 1$. Then $g_y(z) \geq \frac{(z-y^{1/p})^p}{\lambda}$ for $z > y^{1/p}$.*

## 3 Coreset construction

We begin by summarizing some key aspects of the sensitivity framework. Formal definitions for the sensitivity framework (including sensitivities, the VC dimension, and the main subsampling theorem) are given in Appendix A. In the sensitivity framework, the goal is to obtain coresets using importance sampling techniques to approximate loss functions [26]. Given a loss function whose value depends on a collection of input points, the main idea of the framework is to measure the sensitivity of any input point in terms of its worst-case contribution to the loss function. More precisely, given a point $x_j$, its sensitivity for the loss function of the form $f(X\eta) = \sum_{j\in[n]} g(x_j\eta)$ is given by

$$\sigma_j = \sup_\eta \frac{g(x_j\eta)}{f(X\eta)}.$$

The main subsampling theorem, Proposition A.5, combines the sensitivities together with the theory of VC dimension. The idea is to sample points according to probabilities that are proportional to the sensitivities, in order to create an appropriately reweighted subsample of the initial collection of points. Suppose the total sensitivity $S = \sum_{j\in[n]} \sigma_j$ of the points and the VC dimension $\Delta$ associated with a set system based on the summands $g(x_i\eta)$ in the loss function $f(X\eta)$ are bounded, and choose a failure probability $\delta$. If the subsample is of size $k = O(\frac{S}{\varepsilon^2}(\Delta \log(S) + \log(\frac{1}{\delta})))$, then it is in fact a $(1+\varepsilon)$-coreset [19] with probability at least $1 - \delta$. Unfortunately, it is often just as difficult to compute the exact sensitivities as it is to solve the original problem. The remedy is to exploit the fact that one does not need the exact sensitivities themselves: it suffices to use any upper bounds on the

sensitivities, provided that the upper bounds are not too loose, since a larger upper bound will lead to a larger coreset. Thus, we can reduce the task of coreset construction to two tasks: control of the VC dimension and sensitivity estimation of the loss function. This is handled in the following sections.

## 3.1 Bounding the VC dimension

We prove two different bounds on the VC dimension. The first one is a simple quadratic bound of $O(d^2)$. Our proof in the appendix simply counts the number of operations required to compare the loss function to a given threshold. The VC dimension bound then follows from a standard result in the context of bounding the VC dimension of neural networks [4, Thm. 8.14].

**Lemma 3.1.** *The VC dimension of the range space associated with the class of Poisson loss functions as in Equation* (3) *is bounded by $\Delta(\mathfrak{R}_{\mathcal{F}^*}) \leq O(d^2)$.*

It is noteworthy that the quadratic dependence is implied already only from one single application of the exponential function, which is sufficient but likely not necessary in our context. Hence, we show in the remainder a more refined near-linear bound of $O(\frac{d}{\varepsilon} \log(n) \log(y_{\max})) = \tilde{O}(d/\varepsilon)$, while keeping the dependence on other input parameters—namely, on $n$ and $y_{\max}$—logarithmic.

To this end, we subdivide the set of input functions into groups of growing values of their response parameter $y_i$, and of their sensitivity $\varsigma_i$ in a geometric progression. By rounding these values in each group to their next power in the geometric progression, we obtain disjoint sets of functions that closely approximate the original weighted loss functions, and whose VC-dimension can be bounded in $O(d)$. Since there is only a logarithmic number of groups in both progressions, we obtain the claimed VC dimension bound.

Recall the responses $(y_i)_i$ are nonnegative integers. We define their largest value (for a given input) to be $y_{\max} = \max\{y_i \mid i \in [n]\}$. Therefore, they are naturally bounded between $0 \leq y_i \leq y_{\max}$ for all $i \in [n]$ and there are at most $y_{\max} + 1$ different values of $y_i$. Also note that the sensitivity values are naturally bounded by $0 \leq \varsigma_i \leq 1$, but since they are continuous, they must be bounded away from $0$ in order for the geometric progression to end in a finite (logarithmic) number of steps. If we increase each sensitivity by $1/n$ then the total sensitivity grows only by a constant, since $S' = \sum_{i \in [n]} (\varsigma_i + 1/n) = S + 1$. For these reasons, we can thus assume that $1/n \leq \varsigma_i \leq 1$.

Now, we would like to increase the sensitivities even more to their next power of 2, which will clearly increase the total sensitivity by no more than

$$2S' = 2S + 2 \leq 3S. \tag{5}$$

By our above upper and lower bounds on the sensitivities, this will result in $O(\log(n))$ groups, where in each group all sensitivities are equal. Note that in Proposition A.5, the reweighting of points depends only on fixed terms except for the sensitivities. Thus, in each group, all weights are constant. We have the following bound that applies to each group and for any fixed $y \in \mathbb{N}_0$.

**Lemma 3.2.** *The VC dimension of the range space induced by the set of functions $\mathcal{F}_c = \{g_i(\beta) = c \cdot g_{y_i}(x_i\beta) \mid i \in [n]\}$ with equal weight $c \in \mathbb{R}_{\geq 0}$, and equal $y_i = y \in \mathbb{N}_0$ for all $i \in [n]$ satisfies $\Delta(\mathfrak{R}_{\mathcal{F}_c}) = O(d)$.*

For the values of $y_i$, we proceed in a very similar way. However, unlike the sensitivities, a constant approximation as in Equation (5) provided by powers of 2 will not suffice. Instead, we group the values of $y_i$ into powers of $(1 + \varepsilon)$ and round all $y_i$ that belong to the same group to the next larger power. We argue that this preserves a $(1 \pm O(\varepsilon))$ approximation to the original loss function. Indeed, this claim even holds for each summand $g_{y_i}$ if $y_i$ is large enough, as the following lemma shows.

**Lemma 3.3.** *Let $y \geq 8$ and $1 \geq \varepsilon > 0$. Let $y < y' \leq (1 + \varepsilon)y_i$. Then for arbitrary $z > 0$ it holds that $(1 - 3\varepsilon)g_y(z) \leq g_{y'}(z) \leq (1 + 3\varepsilon)g_y(z)$.*

A direct consequence of Lemma 3.3 is that any coreset for the rounded version is a coreset for the original loss function and vice versa, up to an additional $(1 \pm O(\varepsilon))$ error. We can therefore work with the rounded version of the loss function, which yields better bounds for the VC dimension.

A general theorem for bounding the VC dimension of the union or intersection of $t$ range spaces, each of bounded VC dimension at most $D$, was given in [6]. Their result yields $O(tD \log(t))$. Here, we give a bound of $O(tD)$ for the special case that the range spaces are disjoint[2].

---

[2]The same bound and proof also appeared in [20] in a different context.

**Lemma 3.4.** *Let $\mathcal{F}$ be any family of functions, and let $F_1, \ldots, F_t \subseteq \mathcal{F}$ be nonempty sets that form a partition of $\mathcal{F}$, i.e., their disjoint union satisfies $\dot{\bigcup}_{i \in [t]} = \mathcal{F}$. Let the VC dimension of the range space induced by $F_i$ be bounded by $D$ for all $i \in [t]$. Then the VC dimension of the range space induced by $\mathcal{F}$ satisfies $\Delta(\mathfrak{R}_{\mathcal{F}}) \leq tD$.*

As a result of our previous partition into groups and the $O(d)$ bound on each group, we obtain the desired result.

**Lemma 3.5.** *Let $\mathcal{F}$ be the set of functions in the Poisson model. We can round and group the values of $y_i$ and the associated sensitivities $\varsigma_i$ to obtain $\mathcal{F}^*$ such that each function in $\mathcal{F}^*$ is weighted by $0 < w_i \in W := \{u_1, \ldots, u_t\}$ for $t \in O(\varepsilon^{-1} \log(n) \cdot \log(y_{\max}))$. The range space induced by $\mathcal{F}^*$ satisfies $\Delta(\mathfrak{R}_{\mathcal{F}^*}) \leq O(\frac{d}{\varepsilon} \log(n) \log(y_{\max}))$.*

As a direct corollary of Lemmas 3.1 and 3.5, we obtain the following combined bound.

**Corollary 3.6.** *The VC dimension $\Delta(\mathfrak{R}_{\mathcal{F}^*})$ of the range space associated with the class of Poisson loss functions as in Equation (3) is bounded by*

$$\Delta(\mathfrak{R}_{\mathcal{F}^*}) \leq O\left(d \cdot \min\left\{d, \frac{\log(n) \log(y_{\max})}{\varepsilon}\right\}\right).$$

### 3.2 Bounding the sensitivities

We split the loss function into two parts:

$$f_y(X\beta) := \sum_{i: x_i\beta \leq \eta} g_{y_i}(x_i\beta) + \sum_{i: x_i\beta > \eta} g_{y_i}(x_i\beta) \tag{6}$$

We will ignore the first sum, since we will see later in the main approximation of Section 4, that by shifting the hyperplanes defined by parameter vectors in the solution space, everything can be shifted to the second sum where $x_i\beta \geq \eta$. In this way, we preserve a $(1 + \varepsilon)$-approximation, if $\eta = \Theta(\varepsilon)$ is small enough. We note that this shifting technique still requires the extreme points on the convex hull $\mathrm{Ext}(X)$ to be maintained; we address this issue in Section 5. We will focus on bounding the sensitivities for the remaining points with $x_i\beta \geq \eta$. In the next lemma we require the concept of a well-conditioned-basis [12]. Let $q \in \{2, \infty\}$ denote the dual norm of $p \in \{1, 2\}$, respectively. We say that $U$ is a '$(\alpha, \gamma, p)$-well-conditioned basis' for the column span of $X = UR$ if $U \in \mathbb{R}^{n \times d}$ satisfies

$$\|U\|_p \leq \alpha, \quad , \forall z \in \mathbb{R}^d : \|z\|_q \leq \gamma \|Uz\|_p. \tag{7}$$

**Lemma 3.7.** *Let $X \in \mathbb{R}^{n \times d}, y \in \mathbb{N}_0^n$ be a $\rho$-complex dataset, i.e., Equation (2) holds. Let $p \in \{1, 2\}$. Let $\lambda \geq 1$ be the slope parameter from either Lemma 2.2 or Lemma 2.3 depending on the value of $p$. Let $\gamma$ be a conditioning parameter and $\eta > 0$ be arbitrary. Then the sensitivity for each $x_i$ with $x_i\beta > \eta$ is bounded by $\varsigma_i \leq \lambda \rho \gamma^p \|U_i\|_p^p + 2/n$. Their total sensitivity is bounded by $\mathfrak{S} \leq O\left(\rho d \sqrt{y_{\max}/\log(y_{\max})} + \log\log(1/\eta)\right)$ for $p = 1$, and $\mathfrak{S} \leq O(\rho d + \log(y_{\max}) + \log\log(1/\eta))$ for $p = 2$.*

### 3.3 Combining the results into the sensitivity framework

Putting all steps (VC dimension, total sensitivity) together into the sensitivity framework, Proposition A.5 yields the following computational result, where in particular $\ell_p$ well-conditioning is established constructively using $\ell_2$ subspace embeddings [9], resp. using $\ell_1$ spanning sets [44].

**Theorem 3.8.** *Let $X \in \mathbb{R}^{n \times d}, y \in \mathbb{N}_0^n$ be a $\rho$-complex dataset, i.e., Equation (2) holds. We can compute a weighted coreset $(K, w) \in \mathbb{R}^{k \times d} \times \mathbb{R}_{\geq 0}^k$ for the pth-root-link Poisson regression problem with $p \in \{1, 2\}$ on $D(\eta) := \{\beta \in \mathbb{R}^d : \forall i, x_i\beta > \eta\}$. The size of the coreset is bounded by $k = \tilde{O}(\varepsilon^{-2} d \cdot \min\{d, \varepsilon^{-1} \log(n) \log(y_{\max})\} \cdot m)$, where*

$$m = \begin{cases} \rho d \log\log(d) \sqrt{y_{\max}/\log(y_{\max})} + \log\log(1/\eta) & p = 1 \\ \rho d + \log(y_{\max}) + \log\log(1/\eta) & p = 2. \end{cases}$$

## 4 Main approximation result

In the previous section, we developed a coreset for the sum of individual losses where $x_i\beta \geq \eta$, i.e., for the second sum of Equation (6). Since we cannot bound the remaining first sum where $x_i\beta < \eta$, we choose to simply avoid it instead, by shifting each solution by $\eta$. Define for any $\eta \geq 0$

$$D(\eta) \coloneqq \{\beta \in \mathbb{R}^d \ : \ \forall i, \ x_i\beta > \eta\}.^3$$

The original domain of optimization is $D(0)$ and the shifted domain is $D(\eta)$. Shifting the domain does not remove the need to store the extreme points on the convex hull of the input, since we need these points to determine the feasible domain $D(\eta)$ during optimization. However, shifting removes the need to approximate the first sum in Equation (6) over points with unbounded sensitivity located in a small slab of width $\eta$ within the convex hull.

Since we have a $(1 \pm \varepsilon)$ coreset for $D(\eta)$, we need to find a suitable choice for $\eta$ and show that the optimizer $(\beta')^* \in D(\eta)$ is a $(1 + O(\varepsilon))$ approximation for the optimizer in the original domain $\beta^* \in D(0)$. We thus define

$$\tilde{\beta}^* \coloneqq \operatorname{argmin}_{\beta' \in D(\eta)} f(X\beta'), \quad \beta^* \coloneqq \operatorname{argmin}_{\beta \in D(0)} f(X\beta). \tag{8}$$

**Lemma 4.1.** *It holds for sufficiently small $\eta > 0$ that*

$$f(X\beta^*) \leq f(X\tilde{\beta}^*) \leq (1 + O(\eta))f(X\beta^*).$$

Now we combine the preceding results, namely the coreset and the domain shifting bound, for our main theorem.

**Theorem 4.2.** *Let $\varepsilon \in (0, 1/14)$. Let $(C, w)$ be a coreset according to Theorem 3.8. Let $\tilde{\beta} \coloneqq \operatorname{argmin}_{\beta \in D(\varepsilon)} f_w(C\beta)$, $\beta^* \coloneqq \operatorname{argmin}_{\beta \in D(0)} f(X\beta)$. Then*

$$f(X\beta^*) \leq f(X\tilde{\beta}) \leq (1 + \varepsilon)f(X\beta^*).$$

## 5 Extreme points on the convex hull

In the previous sections, we argued that for obtaining a $(1 + \varepsilon)$ approximation, it suffices to calculate a coreset that is valid for all $\beta \in D(\varepsilon)$, and then to minimize our loss function over $\beta \in D(\varepsilon)$ using the coreset instead of the full data. Note that for the optimizer to stay in the feasible set $D(\varepsilon)$, one must store the extreme points on the convex hull of the input points denoted by $\mathrm{Ext}(X)$. This is true even when $\varepsilon = 0$, i.e., even when the original function is considered and no shifting occurs. Note that there exist datasets such as our 'points on a circle' example considered in the lower bounds of Section 6, such that $|\mathrm{Ext}(X)| = n$.

There are several ways to either characterize $|\mathrm{Ext}(X)|$ for *typical* inputs in a sublinear way, or to approximate the convex hull by a smaller sublinear subset, called an $\varepsilon$-kernel, with an error of at most $\varepsilon$. Since these methods are usually relative to the diameter of the data, and since we need an additive error for our shifting approach, we first normalize the data to be within the unit ball. We note that this does not change the value of the loss function, since this involves scaling by a fixed value $C \geq 1$, and since we can use the fact that

$$f(X\beta) = f\left(\left(\frac{X}{C}\right)(C\beta)\right),$$

as well as the one-to-one correspondence between any $\beta$ and $C\beta$. Thus, we can run the algorithm on the rescaled data, obtain a good or optimal solution $C\beta^*$, and rescale $C\beta^*$ to obtain the corresponding $\beta^*$. Rescaling steps such as normalizing data to zero mean and unit variance are standard in statistical data analysis [22].

**Smoothed complexity of the convex hull** Instead of the worst case $|\mathrm{Ext}(X)|$ over all possible datasets $X \in \mathbb{R}^{n \times d}$, in smoothed complexity we consider

$$\sup_{X \in \mathbb{R}^{n \times d}} \mathbb{E}_{\Xi \sim \mathcal{N}} |\mathrm{Ext}(X + \Xi)|,$$

---

[3] Such domain restrictions do not lead to feasibility issues, as discussed in Appendix F due to page limitations.

where $\mathcal{N}$ denotes the distribution over matrices $\Xi$ with the same dimensions as $X$ and with i.i.d. Gaussian entries with mean $0$ and variance $\sigma^2$, i.e., $\Xi_{i,j} \sim N(0, \sigma^2)$ for all $i \in [n], j \in [d]$ [11]. This is motivated by the fact that many datasets are recorded with measurement errors, which can often be assumed to be Gaussian. Specifically for the convex hull, [11, Chapter 4] showed that for normalized data in the unit cube, the supremum above is bounded by

$$O\left(\frac{\log^{\frac{3}{2}d-1}(n)}{\sigma^d} + \log^{d-1}(n)\right),$$

which is sublinear in $n$, though exponential in $d$.

$\varepsilon$**-kernels** The purpose of $\varepsilon$-kernels is to approximate the extent of a point set up to an error of $1 - \varepsilon$ for any direction in $\mathbb{R}^d$ based on a subset of the data. They were introduced by [1] and improved by [8] to optimal $\Theta(1/\varepsilon^{(d-1)/2})$ size, see the survey [2]. Since we assume our data to be normalized within the unit ball, this translates to an additive error that is bounded by $\varepsilon$. Thus, the boundary of the convex hull of the $\varepsilon$-kernel can be smaller than the boundary of the original convex hull by at most $\varepsilon$.

**Improvement for structured data** It is known that $\varepsilon$-kernels can have size up to $\Omega(1/\varepsilon^{(d-1)/2})$ in the worst case. Beyond the worst case, the structure of the given data may allow for much smaller $\varepsilon$-kernels to exist, even in high dimensions. Motivated by this, [5] developed a 'greedy clustering' approach that produces an $\varepsilon$-kernel of size $O(k_{\min}/\varepsilon^2)$, where $k_{\min} = k_{\min}(X, \varepsilon)$ denotes the smallest possible size of a subset that gives the required $\varepsilon$-kernel guarantee for the original input $X$.

**Using $\varepsilon$-kernels for optimizing Poisson models** The above options include the possibility to calculate the convex hull and rely on the sublinear smoothed complexity bound, if this is reasonable in the given context. Otherwise, if we have access to any of the above $\varepsilon$-kernel constructions, we shift the hyperplane away from the approximation of the convex hull, provided by the $\varepsilon$-kernel, by a distance of $\eta = 2\varepsilon$ instead of just $\varepsilon$. As a result, the hyperplane will be shifted away from the original convex hull by at least $\varepsilon$ and at most $2\varepsilon$. Thus, we still compute a $(1 + \Theta(\varepsilon))$ approximation in this way.

## 6 Lower bounds

We complement our coreset constructions for the variants of the Poisson model by a series of lower bounds. Our bounds in Lemma 6.1 and Lemma 6.2 are specifically for the $p$th-root-link, and not for the log-link, which was studied in [31]. We use similar constructions of the bad dataset as those used frequently in previous literature, e.g., in [31, Theorem 6], and in [36, 41, 23, 21]. However, each of these references require specific adaptations to the respective loss function that do not directly apply to our setting. Our arguments are thus adapted to the Poisson $p$th-root-link model to show that it does not admit sublinear coresets without imposing assumptions on the data or restricting the model.

The hard instance consists of $n$ equidistant points on the unit circle. Recall that in the Poisson regression formulation, every point has an additional intercept coordinate which is $1$, and the corresponding parameter of $\beta$ determines the affine translation; see Section 2. For every $i \in [n]$, let $x_i = (1, \cos(\frac{2\pi i}{n}), \sin(\frac{2\pi i}{n}))$, and $y_i = 1$. Recall that any feasible $\beta$ must satisfy $x_i\beta > 0, \forall i \in [n]$. The hyperplane parameterized by $\beta$ is thus always outside the point set and $\beta$ points in the direction of the $(x_i)_{i \in [n]}$.

The idea for showing that any point has sensitivity $1$ is that if $\beta$ points to the center of the point set, and the hyperplane is translated to just 'touch' point $x_i$, then $x_i\beta$ is arbitrarily close to $0$, implying that the cost is arbitrarily large. All other points are sufficiently bounded away from the hyperplane, but also not too far away, so their cost is bounded. This implies that the sensitivity of point $x_i$ can be made arbitrarily close to $1$. By symmetry of our construction, this holds for any point.

**Lemma 6.1.** *Consider a number $n \geq 8$ of points equidistant on a unit circle in a 2-dimensional affine subspace embedded in $\mathbb{R}^d, d \geq 3$, each with label $y_i = 1$. Then the sensitivity of each point for the Poisson model with $p$th-root-link for $p \in \{1, 2\}$ is arbitrarily close to $1$. Consequently, any coreset for the Poisson regression model must comprise all $\Omega(n)$ input points.*

Below, we prove an even more interesting statement, i.e., that no compression is possible below $\Omega(n)$ *bits*. While this statement appears to give a weaker $\Omega(n/\log(n))$ bound against coresets, it in fact gives a stronger bound in some sense. This is because the bound holds against *any* possible data reduction algorithm and against *any* data structure that answers negative log-likelihood queries

to within a small error, independent of what (possibly randomized) operations the data reduction algorithm performs. For example, the algorithm could subsample, it could select input points as coreset constructions do, or it could take linear combinations as in linear sketching. More generally, the bound holds against any sort of bit encoding that represents the reduced data. The reduction is based on the same data example of equidistant points on a circle embedded in $d \geq 3$ dimensions.

**Lemma 6.2.** *Let $\Sigma_D$ be a data structure for $D = [X, y] \in \mathbb{R}^{n \times d} \times \mathbb{R}^n$, $d \geq 3$, that approximates negative log-likelihood queries $\Sigma_D(\beta)$ for Poisson regression with the pth-root-link for $p \in \{1, 2\}$, such that for some $\varphi \geq 1$ it holds that*

$$\forall \beta \in \mathbb{R}^d : f(X\beta) \leq \Sigma_D(\beta) \leq \varphi \cdot f(X\beta).$$

*If $\varphi < \frac{n}{8\log(n)}$ then $\Sigma_D$ requires $\Omega(n)$ bits of memory.*

Next, we prove that the parts of our analysis that are specific to the Poisson model are tight. In particular, the scale parameter $\lambda$ is only a constant for $p = 2$, but in the case $p = 1$ we only have a $\sqrt{y/\log(y)}$ upper bound. Since $y_{\max}$, i.e., the largest $y$, can potentially be very large, one may ask if we can do better. Our next result exactly characterizes the smallest possible parameter $\lambda$ such that our linear lower envelope approximation is tangent to the actual loss function, in order to show a tight $\lambda = \Theta(\sqrt{y/\log(y)})$ bound. This characterization of tangent points relies on properties of the principal branch $W_0$ of the Lambert function, which is defined by the equation

$$W_0(x)e^{W_0(x)} = x, \text{ for } x \geq -1/e.$$

The only approximations that the final bounds obey follow from Stirling's approximation and from the following upper bound on the Lambert function $W_0$ by a square root function. This upper bound improves the recent cubic root bound of [38, Theorem 3.2] within a small region, and is crucial to obtaining a tight square root (rather than cubic root) lower bound on $\lambda$.

**Lemma 6.3.** *For all $x \in [-1/e, 0)$, it holds that $W_0(x) \leq \sqrt{2(1 + ex)} - 1$.*

The above bound on the $W_0$ function is novel and may be of independent interest. In our context, it allows us to prove the following tight bound on $\lambda$ that resembles the same asymptotic upper bound as in Lemma 2.2, and establishes a matching lower bound.

**Lemma 6.4.** *Let $y \in \mathbb{N}$ be arbitrary, $p = 1$, and $\tau = y^{1/p}$ in the definition Equation (4) of $g_y$. Let $h_\lambda(z) := \frac{(z - y^{1/p})^p}{\lambda}$ for $z > 0$. Then $g_y$ and $h_\lambda$ are tangent to each other if and only if $\lambda = \lambda^*(y) = (W_0(\frac{-y}{(y!)^{1/y}\exp(2)}) + 1)^{-1}$, in which case the unique tangent point is $z^*(y) = \frac{y\lambda^*(y)}{\lambda^*(y) - 1}$. In addition, $\lambda^*(y) = \Theta(\sqrt{y_{\max}/\log(y_{\max})})$.*

The next lemma shows for $p \geq 3$ that there exists no constant $C$ such that the domain shifting approach that we developed in Section 4 yields a $1 + C\varepsilon$ error bound. As this error bound is a crucial sufficient condition for our main approximation results given in Lemma 4.1 and Theorem 4.2 to hold, the lemma suggests that different techniques may be needed. It indicates a limitation of our analysis to the most common values $p \in \{1, 2\}$, which are the main parameterizations considered in our work.

**Lemma 6.5.** *Let $p \in \mathbb{N}$, $p \geq 3$. Then there does not exist an absolute constant $C \geq 0$ such that for all sufficiently small $\eta > 0$ and for all $\beta \in D(0)$, $\beta' := \beta + \eta e_1 \in D(\eta)$ satisfies*

$$f(X\beta') \leq f(X\beta) + \eta^p n + \eta C f(X\beta). \tag{9}$$

## 7 Concluding remarks

In Section 1.3, we recalled that previous finite sample size results had either unbounded or $O(\sqrt{d})$ error instead of our $(1 + \varepsilon)$ approximation. Our lower bounds on the parameters, together with linear VC dimension, linear sensitivity, and linear $\rho$ dependence in our main quantitative bounds of Theorem 3.8, leave no room for improvement (up to polylogarithmic factors) if one uses a black-box application of the sensitivity framework. We remark that recent improvements on $\ell_p$ sensitivity sampling [33] suggest that the dimension dependence can be improved to linear as well. Exploiting the fact that our coreset gives a guarantee for all $\beta \in D(\varepsilon)$, it would be interesting to extend the statistical treatment to the Bayesian setting, inferring the distribution of parameters over this (sub-)domain, as was recently accomplished in the case of logistic and probit regression [14].

## Acknowledgments and Disclosure of Funding

The authors thank the reviewers for their constructive feedback and discussions. We also thank Simon Omlor for valuable discussions that inspired the domain shift idea and Tim Novak for help with experiments. The research of HCL has been partially funded by the DFG — Project-ID 318763901 — SFB1294. AM was mainly supported by the German Research Foundation (DFG) — grant MU 4662/2-1 (535889065) and by the TU Dortmund - Center for Data Science and Simulation (DoDaS). AM acknowledges additional travel funding by the University of Cologne.

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

## A  Details on the Sensitivity Framework

**Definition A.1** (Coreset, cf. [19]). *Let $X \in \mathbb{R}^{n \times d}$ be a set of points $\{x_1, \ldots, x_n\}$, weighted by $w \in \mathbb{R}_{>0}^n$. For any $\eta \in \mathbb{R}^d$, let the cost of $\eta$ w.r.t. the point $x_i$ be described by a function $w_i \cdot f(x_i \eta)$ mapping from $\mathbb{R}$ to $(0, \infty)$. Thus, the cost of $\eta$ w.r.t. the (weighted) set $X$ is $f_w(X \eta) = \sum_i w_i \cdot f(w_i \eta)$. Then a set $K \in \mathbb{R}^{k \times d}$, (re)weighted by $u \in \mathbb{R}_{>0}^k$ is a $(1 + \varepsilon)$-coreset of $X$ for the function $f_w$ if $k \ll n$ and*

$$\forall \beta \in \mathbb{R}^d \colon |f_w(X\eta) - f_u(K\eta)| \le \varepsilon \cdot f_w(X\eta).$$

In our analysis we use sampling based on so-called sensitivity scores, the range space induced by the set of functions, and the VC-dimension. We define these notions next.

**Definition A.2** (Sensitivity, [26]). *Consider a family of functions $\mathcal{F} = \{g_1, \ldots, g_n\}$ mapping from $\mathbb{R}^d$ to $[0, \infty)$ and weighted by $w \in \mathbb{R}_{>0}^n$. The sensitivity of $g_\ell$ for the function $f_w(\eta) = \sum_{\ell \in [n]} w_\ell g_\ell(\eta)$, where $\eta \in \mathbb{R}^d$, is*

$$\varsigma_\ell = \sup \frac{w_\ell g_\ell(\eta)}{f_w(\eta)}. \tag{10}$$

*The total sensitivity is $\mathfrak{S} = \sum_{\ell \in [n]} \varsigma_\ell$.*

**Definition A.3** (Range space; VC dimension). *A range space is a pair $\mathfrak{R} = (\mathcal{F}, \mathtt{ranges})$, where $\mathcal{F}$ is a set and $\mathtt{ranges}$ is a family of subsets of $\mathcal{F}$. The VC dimension $\Delta(\mathfrak{R})$ of $\mathfrak{R}$ is the size $|G|$ of the largest subset $G \subseteq \mathcal{F}$ such that $G$ is shattered by $\mathtt{ranges}$, i.e., $|\{G \cap R : R \in \mathtt{ranges}\}| = 2^{|G|}$.*

**Definition A.4** (Induced range space). *Let $\mathcal{F}$ be a finite set of functions mapping from $\mathbb{R}^d$ to $\mathbb{R}_{\ge 0}$. For every $x \in \mathbb{R}^d$ and $r \in \mathbb{R}_{\ge 0}$, let $\mathtt{range}_{\mathcal{F}}(x, r) = \{f \in \mathcal{F} : f(x) \ge r\}$, and $\mathtt{ranges}(\mathcal{F}) = \{\mathtt{range}_{\mathcal{F}}(x, r) : x \in \mathbb{R}^d, r \in \mathbb{R}_{\ge 0}\}$. Let $\mathfrak{R}_{\mathcal{F}} = (\mathcal{F}, \mathtt{ranges}(\mathcal{F}))$ be the range space induced by $\mathcal{F}$.*

To construct coresets for Poisson models, we use a framework that combines sensitivity scores with the theory of VC dimension, originally proposed by [18, 7]. We use a more recent and slightly updated version as stated in the following theorem.

**Proposition A.5** ([19], Theorem 31). *Consider a family of functions $\mathcal{F} = \{f_1, \ldots, f_n\}$ mapping from $\mathbb{R}^d$ to $[0, \infty)$ and a vector of weights $w \in \mathbb{R}_{>0}^n$. Let $\varepsilon, \delta \in (0, 1/2)$. Let $s_i \ge \varsigma_i$. Let $S = \sum_{i=1}^n s_i \ge \mathfrak{S}$. Given $s_i$ one can compute in time $O(|\mathcal{F}|)$ a set $\mathcal{R} \subset \mathcal{F}$ of*

$$O\left(\frac{S}{\varepsilon^2}\left(\Delta \log(S) + \log\left(\frac{1}{\delta}\right)\right)\right)$$

*weighted functions such that with probability $1 - \delta$ we have for all $\eta \in \mathbb{R}^d$ simultaneously*

$$\left| \sum_{f \in \mathcal{F}} w_i f_i(\eta) - \sum_{f \in \mathcal{R}} u_i f_i(\eta) \right| \le \varepsilon \sum_{f \in \mathcal{F}} w_i f_i(\eta),$$

*where each element of $\mathcal{R}$ is sampled i.i.d. with probability $p_j = \frac{s_j}{S}$ from $\mathcal{F}$, $u_i = \frac{S w_j}{|\mathcal{R}| s_j}$ denotes the weight of a function $f_i \in \mathcal{R}$ that corresponds to $f_j \in \mathcal{F}$, and where $\Delta$ is an upper bound on the VC dimension of the range space $\mathfrak{R}_{\mathcal{F}^*}$ induced by $\mathcal{F}^*$ that can be obtained by defining $\mathcal{F}^*$ to be the set of functions $f_j \in \mathcal{F}$ where each function is scaled by $\frac{S w_j}{|\mathcal{R}| s_j}$.*

## B  Proofs for Poisson $p$th-root-link model

**Lemma 2.1.** *It holds for all $z \in \mathbb{R}_{>0}, p \in [1, \infty), y \in \mathbb{N}$ that*

$$g_y(z) = z^p - py \log(z) + \log(y!) \ge \max\left\{1, \frac{1}{3}(1 + p\log(z))\right\}.$$

*Proof of Lemma 2.1.* For $y = 1$ a direct calculation yields $g_y(z) \ge g_y(y^{1/p}) = 1$.

For all other $y \in \mathbb{N} \setminus \{1\}$ we get from Stirling's approximation

$$\log(y!) \geq y \log(y) - y + \frac{1}{2} \log(2\pi y) + \frac{1}{12y + 1}$$

$$\geq y \log(y) - y + \frac{1}{2} \log(y) + \frac{1}{2} \log(2\pi)$$

$$\geq y \log(y) - y + \frac{1}{2} \log(y) + \frac{9}{10}.$$

Since $g_y$ has a unique minimizer at $z^* = y^{1/p}$, it follows that

$$g_y(z) \geq g_y(y^{1/p}) = y - y \log(y) + \log(y!) \geq \frac{9}{10} + \frac{1}{2} \log(y) > 1.$$

To prove the claimed bound generalizing to all $z \in \mathbb{R}_{>0}$, we first argue that

$$\frac{9}{10} + \frac{1}{3} \log(y) \geq \frac{1}{3}(1 + \log(y + 1)), \tag{11}$$

which is equivalent to showing for arbitrary $y \in \mathbb{N}$ it holds that

$$\frac{1}{3}\left(1 + \log\left(\frac{y + 1}{y}\right)\right) = \frac{1}{3}\left(1 + \log\left(1 + \frac{1}{y}\right)\right) \leq \frac{1}{3} + \frac{1}{3y} \leq \frac{2}{3} \leq \frac{9}{10},$$

where the first inequality follows from the inequality $1 + x \leq e^x$ for all values of $x$. We see that by monotonicity of the functions $h(z) = \frac{1}{3}(1 + \log(z^p))$ and the above properties of $g_y(z)$, we have for all $z \leq (y + 1)^{1/p}$ that

$$g_y(z) \geq \frac{9}{10} + \frac{1}{2} \log(y) \geq \frac{9}{10} + \frac{1}{3} \log(y) \overset{\text{Equation (11)}}{\geq} \frac{1}{3}(1 + \log(y + 1)) \geq h(z),$$

where the first inequality above follows from the inequality one line above Equation (11). Finally for $z \geq (y + 1)^{1/p}$ the function $g_y$ grows at least as fast as the lower bound, since

$$g'_y(z) = pz^{p-1} - \frac{py}{z} = p\left(z^{p-1} - \frac{y}{z}\right) = p\left(\frac{z^p - y}{z}\right) \geq p\frac{y + 1 - y}{z} \geq \frac{p}{3z} = h'(z).$$

$\square$

**Lemma 2.2.** *For any $p \geq 1$ and $y \in \mathbb{N}$ it holds that $z^p \geq g_y(z)$ for $z > y^{1/p}$. If $p = 1$, then for some $\lambda \in O(\sqrt{y / \log(y)})$, it holds that $g_y(z) \geq \frac{(z - y^{1/p})^p}{\lambda}$ for $z > y^{1/p}$.*

*Proof of Lemma 2.2.* We start with the upper bound. Using the fact that the assumption $z > y^{1/p}$ implies $\log z > 0$, and setting $y_i = 1$ in Equation (4) yields

$$g_1(z) = z^p - p \log(z) + \log 1 < z^p.$$

It remains to prove the claim for $y \geq 2$. From Stirling's approximation we get

$$\log(y!) \leq y \log(y) - y + \frac{1}{2} \log(y) + \frac{1}{2} \log(2\pi) + \frac{1}{12}.$$

Now, note that since the derivative of the function $y \mapsto \frac{1}{2} \log y - y$ is strictly negative over the interval $[2, \infty)$, it follows that the function itself is decreasing over the same interval. By rearranging terms and replacing $y_i$ in Equation (4) with an arbitrary $y \geq 2$, it follows that

$$g_y(z) = z^p - y \log(z^p) + \log(y!) \leq z^p - y \log(y) + \log(y!)$$

$$\leq z^p + \frac{1}{2} \log(y) - y + \frac{1}{2} \log(2\pi) + \frac{1}{12}$$

$$\leq z^p + \frac{1}{2} \log(2) - 2 + \frac{1}{2} \log(2\pi) + \frac{1}{12}$$

$$= z^p + \frac{1}{2} \log(\pi) - \frac{23}{12} < z^p,$$

where the last inequality holds since $\pi < e^2$, and thus $\frac{1}{2}\log(\pi) < \log(e) = 1$.

In the remainder, let $p = 1$. Let

$$LB(y) := \max\left\{1, \frac{1}{3}\left(1 + \log(y)\right)\right\}$$

be the lower bound given in Lemma 2.1. Now, we want to prove that $g_y(z) \geq \frac{z-y}{\lambda} =: h(z)$ for $\lambda \geq 1$ as small as possible.

To this end, we consider the derivatives $g_y'(z) = 1 - \frac{y}{z}$ and $h'(z) = \frac{1}{\lambda}$, and find that

$$1 - \frac{y}{z} \geq \frac{1}{\lambda} \iff z \geq y\left(1 + \frac{1}{\lambda - 1}\right).$$

Next, we see that since $h(y) = 0$, we can guarantee $h(z) \leq LB(y) \leq g_y(z)$ for all $z \in [y, y + \Delta]$, where $\Delta := \lambda \cdot LB(y)$.

To obtain a general lower bound on $g_y(z)$, we want both conditions to hold simultaneously, which is true whenever

$$\frac{y}{\lambda - 1} \leq \Delta \iff y \leq \lambda(\lambda - 1)LB(y)$$

Solving for $\lambda$ yields that $g_y(z) \geq \frac{z-y}{\lambda}$ holds for all $\lambda \geq \frac{1}{2}\left(\sqrt{\frac{4y}{LB(y)} + 1} + 1\right)$. $\qquad\square$

**Lemma 2.3.** *Let $p \geq 2$ and $y \in \mathbb{N}$, and $\lambda = 1$. Then $g_y(z) \geq \frac{(z - y^{1/p})^p}{\lambda}$ for $z > y^{1/p}$.*

*Proof of Lemma 2.3.* First we define for $\tau > 0$ the function $h_\lambda$, and compute its derivatives:

$$h_\lambda(z) := \frac{(z - \tau)^p}{\lambda}, \quad h_\lambda'(z) = p\frac{(z - \tau)^{p-1}}{\lambda}, \quad h_\lambda''(z) = p(p - 1)\frac{(z - \tau)^{p-2}}{\lambda}, \quad z \in \mathbb{R}_{>0}. \quad (12)$$

Setting $\tau = y^{1/p}$ in Equation (12), we obtain

$$h_\lambda(y^{1/p}) = \frac{(y^{1/p} - y^{1/p})^p}{\lambda} = 0 < \frac{1}{2}\log(2\pi y) < g_y(y^{1/p}),$$

where the second inequality follows from the fact that the minimizer of $g_y$ is $y^{1/p}$ and from Stirling's approximation:

$$\frac{1}{2}\log(2\pi y) + \frac{1}{12y + 1} < \min_{z > 0} g_y(z) = y - y\log y + \log y!.$$

Thus, $g_y(y^{1/p}) \geq h_\lambda(y^{1/p})$, and a sufficient condition for $g_y(z) \geq h_\lambda(z)$ to hold for $z > y^{1/p}$ is that $g_y(z) - h(z)$ is nondecreasing on $z > y^{1/p}$, i.e.

$$0 \leq g_y'(z) - h'(z) = p\left[\left(z^{p-1} - \frac{y}{z}\right) - \frac{(z - y^{1/p})^{p-1}}{\lambda}\right] \iff z^{p-1} \geq \frac{y}{z} + \frac{(z - y^{1/p})^{p-1}}{\lambda}.$$

Note that

$$z > y^{1/p} \implies \frac{y}{z} + \frac{(z - y^{1/p})^{p-1}}{\lambda} \leq \frac{y}{y^{1/p}} + \frac{(z - y^{1/p})^{p-1}}{\lambda},$$

so a sufficient condition for $g_y'(z) - h_\lambda'(z)$ to be nonnegative for all $z > y^{1/p}$ is

$$z^{p-1} \geq \frac{(z - y^{1/p})^{p-1}}{\lambda} + (y^{1/p})^{p-1}, \quad \forall z > y^{1/p}.$$

If $\lambda \geq 1$, then a further sufficient condition for $g_y'(z) - h_\lambda'(z)$ to be nonnegative for all $z > y^{1/p}$ is

$$z^{p-1} \geq (z - y^{1/p})^{p-1} + (y^{1/p})^{p-1}, \quad \forall z > y^{1/p},$$

which is equivalent to

$$\left(\frac{z}{y^{1/p}}\right)^{p-1} \geq \left(\frac{z}{y^{1/p}} - 1\right)^{p-1} + 1, \quad \forall z > y^{1/p}. \quad (13)$$

For $x \in \mathbb{R}$, let $\lfloor x \rfloor := \sup\{p \in \mathbb{Z} \ : \ p \le x\}$ be the floor of $x$. We will prove that Equation (13) holds. We have

$$\left(\frac{z}{y^{1/p}}\right)^{p-1}$$

$$= \left(\frac{z}{y^{1/p}}\right)^{p-1-\lfloor p-1 \rfloor} \left(\frac{z}{y^{1/p}}\right)^{\lfloor p-1 \rfloor}$$

$$= \left(\frac{z}{y^{1/p}}\right)^{p-1-\lfloor p-1 \rfloor} \left(\frac{z}{y^{1/p}} - 1 + 1\right)^{\lfloor p-1 \rfloor}$$

$$= \left(\frac{z}{y^{1/p}}\right)^{p-1-\lfloor p-1 \rfloor} \sum_{m=0}^{\lfloor p-1 \rfloor} \binom{\lfloor p-1 \rfloor}{m} \left(\frac{z}{y^{1/p}} - 1\right)^{m} \qquad \text{binomial thm., } p \ge 2$$

$$\ge \left(\frac{z}{y^{1/p}}\right)^{p-1-\lfloor p-1 \rfloor} \left(\left(\frac{z}{y^{1/p}} - 1\right)^{\lfloor p-1 \rfloor} + 1\right) \qquad \frac{z}{y^{1/p}} > 1$$

$$= \left(\frac{z}{y^{1/p}}\right)^{p-1-\lfloor p-1 \rfloor} \left(\frac{z}{y^{1/p}} - 1\right)^{\lfloor p-1 \rfloor} + \left(\frac{z}{y^{1/p}}\right)^{p-1-\lfloor p-1 \rfloor}$$

$$\ge \left(\frac{z}{y^{1/p}} - 1\right)^{p-1-\lfloor p-1 \rfloor} \left(\frac{z}{y^{1/p}} - 1\right)^{\lfloor p-1 \rfloor} + \left(\frac{z}{y^{1/p}}\right)^{p-1-\lfloor p-1 \rfloor} \qquad r \ge 0 \Rightarrow \frac{\mathrm{d}}{\mathrm{d}x}x^r \ge 0$$

$$> \left(\frac{z}{y^{1/p}} - 1\right)^{p-1} + 1 \qquad \frac{z}{y^{1/p}} > 1,$$

as desired. $\qquad \qquad \square$

## C  Proofs for coreset construction

**Lemma 3.1.** *The VC dimension of the range space associated with the class of Poisson loss functions as in Equation (3) is bounded by $\Delta(\mathfrak{R}_{\mathcal{F}^*}) \le O(d^2)$.*

*Proof of Lemma 3.1.* Because we use the coreset approach, we need to consider *weighted* subsets of the data, and thus weighted sums $\sum_{i=1}^{n} w_i g_{y_i}(x_i\beta)$ that allow for different weights in Equation (3). In determining the VC dimension we range over all $r \in \mathbb{R}, \beta \in \mathbb{R}^d$, and need to check whether $w_i g_{y_i}(x_i\beta) > r$. Note that the log factorial terms are independent of $x_i\beta$. In particular, for each $r \in \mathbb{R}$, there exists $\mathbb{R} \ni s := r - w_i \log(y_i!)$. We can thus instead count the number of operations required for evaluating each summand

$$w_i((x_i\beta)^p - py_i \ln(x_i\beta)) \ge s.$$

This can be rearranged to

$$x_i\beta \le \exp\left(\frac{(x_i\beta)^p - s/w_i}{py_i}\right)$$

which can be accomplished using (at most) the following operations

- 1 division to compute $s/w_i$
- $d$ multiplications and $d - 1$ additions to compute $x_i\beta$
- $p - 1$ multiplications of $x_i\beta$ to compute $(x_i\beta)^p$
- 1 subtraction to compute $(x_i\beta)^p - s/w_i$
- 1 multiplication to compute $py_i$ (0 for $p = 1$)
- 1 division by $py_i$
- 1 exponential function evaluation to compute $\exp\left(\frac{(x_i\beta)^p - s/w_i}{py_i}\right)$
- 1 operation to verify whether $x_i\beta \le \exp\left(\frac{(x_i\beta)^p - s/w_i}{py_i}\right)$ holds

- 1 operation to output 0 or 1 depending on whether $x_i\beta \leq \exp\left(\frac{(x_i\beta)^p - s/w_i}{py_i}\right)$ holds

for a total of $t = 2d + p + 5$ operations, exactly $q = 1$ of which is an exponential function evaluation. Putting these quantities into the final conclusion of [4, Thm. 8.14] yields

$$\Delta(\mathfrak{R}_{\mathcal{F}^*}) \leq (d(q+1))^2 + 11d(q+1)(t + \log_2((9d(q+1))))$$
$$= (2d)^2 + 22d(2d + p + 5 + \log_2(18d)) = O(d^2).$$

$\square$

**Lemma 3.2.** *The VC dimension of the range space induced by the set of functions $\mathcal{F}_c = \{g_i(\beta) = c \cdot g_{y_i}(x_i\beta) \mid i \in [n]\}$ with equal weight $c \in \mathbb{R}_{\geq 0}$, and equal $y_i = y \in \mathbb{N}_0$ for all $i \in [n]$ satisfies $\Delta(\mathfrak{R}_{\mathcal{F}_c}) = O(d)$.*

*Proof of Lemma 3.2.* We split the functions $g_i$ into subfunctions $g_{\leq y_i^{1/p}}(x_i\beta)$ and $g_{> y_i^{1/p}}(x_i\beta)$ depending on whether $x_i\beta \leq y_i^{1/p}$ or $x_i\beta > y_i^{1/p}$. Since $g_i$ is minimized when $x_i\beta = y_i^{1/p}$, and due to strict convexity, the two subfunctions $g_{\leq y_i^{1/p}}\colon (0, y_i^{1/p}] \to [g(y_i^{1/p}), \infty)$ and $g_{> y_i^{1/p}}\colon (y_i^{1/p}, \infty) \to [g(y_i^{1/p}), \infty)$ are strictly monotonic and invertible on their respective ranges and domains.

Now fix an arbitrary subset $G \subseteq \mathcal{F}_c$. Let $\Omega = \mathbb{R}^d \times \mathbb{R}_{\geq 0}$. We have the following bound:

$$|\{G \cap R \mid R \in \texttt{ranges}(\mathcal{F}_c)\}| = |\{\texttt{range}_G(\beta, r) \mid \beta \in \mathbb{R}^d, r \in \mathbb{R}_{\geq 0}\}|$$

$$= \left| \bigcup_{(\beta, r) \in \Omega} \{\{g_i \in G \mid g_i(\beta) \geq r\}\} \right|$$

$$= \left| \bigcup_{(\beta, r) \in \Omega} \{\{g_i \in G \mid c \cdot g_{\leq y_i^{1/p}}(x_i\beta) \geq r \vee c \cdot g_{> y_i^{1/p}}(x_i\beta) \geq r\}\} \right|$$

$$\leq \left| \bigcup_{(\beta, r) \in \Omega} \{\{g_i \in G \mid x_i\beta \geq g_{\leq y_i^{1/p}}^{-1}(r/c)\}\} \right| \cdot \left| \bigcup_{(\beta, r) \in \Omega} \{\{g_i \in G \mid x_i\beta \geq g_{> y_i^{1/p}}^{-1}(r/c)\}\} \right|. \quad (14)$$

The inequality holds, since each non-empty set in the collection on the LHS satisfies either of the conditions of the sets in the collections on the RHS, or both, and is thus the union of two of the sets, one from each collection. It can thus comprise at most all unions obtained from combining any two of these sets.

Now, note that both sets are of the form $\{g_i \in G \mid x_i\beta \geq s_1\}$ where $s_1 = g_{\leq y_i^{1/p}}^{-1}(r/c)$ maps any real $r$ to a value of some subset of the reals $s \in D \subset \mathbb{R}$ as specified above. Extending the domain of $s$ and $x_i\beta$ to the reals, we obtain exactly the points that are shattered by the affine hyperplane classifier $x_i \mapsto \mathbf{1}_{\{x_i\beta - s \geq 0\}}$. The VC dimension of the set of hyperplane classifiers is $d + 1$ [24, 42]. The argument holds verbatim for $s_2 = g_{> y_i^{1/p}}^{-1}(r/c)$.

We conclude the claimed bound on $\Delta(\mathfrak{R}_{\mathcal{F}_c})$ by showing that the above term Equation (14) is strictly less than $2^{|G|}$ for $|G| = 10(d + 1)$. By a bound on the growth of the sets (see [6, 24]), we have for this particular choice

$$\left| \bigcup_{(\beta, r) \in \Omega} \{\{g_i \in G \mid x_i\beta \geq g_{\leq y_i^{1/p}}^{-1}(r/c)\}\} \right| \cdot \left| \bigcup_{(\beta, r) \in \Omega} \{\{g_i \in G \mid x_i\beta \geq g_{> y_i^{1/p}}^{-1}(r/c)\}\} \right|$$

$$\leq \left| \{\{g_i \in G \mid x_i\beta - s_1 \geq 0\} \mid \beta \in \mathbb{R}^d, s_1 \in \mathbb{R}\} \right| \cdot \left| \{\{g_i \in G \mid x_i\beta - s_2 \geq 0\} \mid \beta \in \mathbb{R}^d, s_2 \in \mathbb{R}\} \right|$$

$$\leq \left( \left(\frac{e|G|}{d+1}\right)^{(d+1)} \right)^2 < \left( 30^{(d+1)} \right)^2 = 2^{2(d+1)\log_2(30)} \leq 2^{10(d+1)} = 2^{|G|}$$

which implies that $\Delta(\mathfrak{R}_{\mathcal{F}_{\ell_1}}) < 10(d + 1)$. $\square$

**Lemma 3.3.** *Let $y \geq 8$ and $1 \geq \varepsilon > 0$. Let $y < y' \leq (1+\varepsilon)y_i$. Then for arbitrary $z > 0$ it holds that $(1 - 3\varepsilon)g_y(z) \leq g_{y'}(z) \leq (1 + 3\varepsilon)g_y(z)$.*

*Proof of Lemma 3.3.* Recall that

$$g_{y'}(z) = \underbrace{(z)^p - py' \log(z)}_{(i)} + \underbrace{\log(y'!)}_{(ii)}.$$

We bound the two parts separately. The first part $(i)$ is straightforward from the assumption on $y'$. We only need to distinguish two cases, depending on which either the upper or the lower bound of the assumption $y < y' \leq (1+\varepsilon)y_i$ applies: If $\log(z) \geq 0$, then there is nothing to prove since in that case

$$z^p - py' \log(z) \leq z^p - py \log(z),$$

given the hypothesis that $y < y'$. Suppose that $\log(z) < 0$. Then

$$z^p - py' \log(z) \leq z^p - (1+\varepsilon)py \log(z) \leq (1+3\varepsilon)(z^p - py \log(z)),$$

where the first inequality follows from the hypothesis that $y' \leq (1+\varepsilon)y_i$ and the second inequality follows since $1 + 3\varepsilon > 1 + \varepsilon > 1$.

For the second part $(ii)$ we need some technical claims. Since $y, y' \in \mathbb{N}_0$, it must hold that $y' \geq y+1$. Then

$$\frac{1}{12y'} \leq \frac{1}{12(y+1)} \leq \frac{1}{12y+1}.$$

Further, for any $x \geq e$ it holds that

$$\log((1+\varepsilon)x) = \log(x) + \log(1+\varepsilon) \leq \log(x) + \varepsilon \cdot 1 \leq (1+\varepsilon)\log(x),$$

where the last inequality uses the condition that $x \geq e$.

With this in place, and using $y' > y \geq 8 > e^2$, we apply Stirling's approximation with a constant $C := \frac{1}{2}\log(2\pi)$ that is independent of $y, y'$, and that is common to both the upper and lower bounds from Stirling's approximation for the factorial function. We thus obtain

$$\log(y'!) \leq y' \log(y') - y' + \frac{1}{2}\log(y') + \frac{1}{12y'} + C$$

$$= y'(\log(y') - 1) + \frac{1}{2}\log(y') + \frac{1}{12y'} + C$$

$$= y'(\log(y'/e)) + \frac{1}{2}\log(y') + \frac{1}{12y'} + C$$

$$\leq (1+\varepsilon)^2 y \log(y/e) + (1+\varepsilon)\frac{1}{2}\log(y) + \frac{1}{12y+1} + C$$

$$\leq (1+\varepsilon)^2 \left( y \log(y/e) + \frac{1}{2}\log(y) + \frac{1}{12y+1} + C \right)$$

$$\leq (1+3\varepsilon)\log(y!)$$

where the first inequality follows from the upper bound in Stirling's approximation for the factorial, the second inequality follows from the hypothesis that $y' \leq (1+\varepsilon)y$ and the inequality $\frac{1}{12y'} \leq \frac{1}{12y+1}$ proven above, and the last inequality follows from the lower bound in Stirling's approximation of the factorial and from the fact that $(1+\varepsilon)^2 = 1 + 2\varepsilon + \varepsilon^2 \leq 1 + 3\varepsilon$. The lower bound can be treated in a similar way. Overall, our claim follows. $\square$

**Lemma 3.4.** *Let $\mathcal{F}$ be any family of functions, and let $F_1, \ldots, F_t \subseteq \mathcal{F}$ be nonempty sets that form a partition of $\mathcal{F}$, i.e., their disjoint union satisfies $\dot{\bigcup}_{i \in [t]} = \mathcal{F}$. Let the VC dimension of the range space induced by $F_i$ be bounded by $D$ for all $i \in [t]$. Then the VC dimension of the range space induced by $\mathcal{F}$ satisfies $\Delta(\mathfrak{R}_{\mathcal{F}}) \leq tD$.*

*Proof of Lemma 3.4.* We prove the claim by contradiction. To this end suppose the VC dimension for $\mathcal{F}$ is strictly larger than $tD$. Then there exists a set $G$ of size $|G| > tD$ that is shattered by the ranges of $\mathfrak{R}_\mathcal{G}$. Consider its intersections $G_i = G \cap F_i, i \in [t]$ with the sets $F_i$. By their disjointness, each $G_i$ must be shattered by the ranges of $\mathfrak{R}_{F_i}$. Note, that at least one $G_i$ must therefore satisfy $|G_i|/t > D$, which contradicts the assumption that their VC dimension is bounded by $D$. Our claim thus follows. $\qquad\square$

**Lemma 3.5.** *Let $\mathcal{F}$ be the set of functions in the Poisson model. We can round and group the values of $y_i$ and the associated sensitivities $\varsigma_i$ to obtain $\mathcal{F}^*$ such that each function in $\mathcal{F}^*$ is weighted by $0 < w_i \in W := \{u_1, \ldots, u_t\}$ for $t \in O(\varepsilon^{-1} \log(n) \cdot \log(y_{\max}))$. The range space induced by $\mathcal{F}^*$ satisfies $\Delta(\mathfrak{R}_{\mathcal{F}^*}) \leq O(\frac{d}{\varepsilon} \log(n) \log(y_{\max}))$.*

*Proof of Lemma 3.5.* We partition our input functions $g_{y_i}, i \in [n]$ into disjoint sets with boundaries that increase in a geometric progression, depending on the sensitivities resp. weights, and on the response values

$$G_{ij} = \{g_{y_k} \mid 8 \cdot (1 + \varepsilon)^i \leq y_k < \lfloor 8 \cdot (1 + \varepsilon)^{i+1} \rfloor, 2^j \varsigma_{\min} \leq \varsigma_k < 2^{j+1} \varsigma_{\min}\},$$
$$i \in [0, O(\log_{1+\varepsilon}(y_{\max}))], j \in [0, O(\log(n))].$$

Additionally, we put the remaining values of $y$ into a constant number of disjoint sets

$$H_{yj} = \{g_{y_k} \mid y_k = y, 2^j \varsigma_{\min} \leq \varsigma_k < 2^{j+1} \varsigma_{\min}\}, y \in \{0, 1, 2, \ldots, 7\}, j \in [0, O(\log(n))].$$

In particular, we note that

$$\mathcal{F} = \left(\dot{\bigcup}_{ij} G_{ij}\right) \dot{\cup} \left(\dot{\bigcup}_{yj} H_{yj}\right)$$

forms a partition of the whole function family, since the sets are disjoint and cover all functions by construction.

Each member of a set is of the same form, i.e., after rounding the weights and $y_i$, all members of a subset have equal $y_i = y$, and they have equal weight. The assumptions are thus satisfied for invoking Lemma 3.2 to bound the VC dimension for each of the induced range spaces by $O(d)$. By construction, the subsets are disjoint and their number is bounded by

$$t = O(\log_2(n) \cdot \log_{1+\varepsilon}(y_{\max})) = O\left(\log(n) \cdot \frac{\log(y_{\max})}{\log(1+\varepsilon)}\right) = O(\varepsilon^{-1} \log(n) \log(y_{\max})).$$

We can thus invoke Lemma 3.4 to obtain

$$\Delta(\mathfrak{R}_{\mathcal{F}^*}) \leq O(dt) = O\left(\frac{d}{\varepsilon} \log(n) \log(y_{\max})\right) = \tilde{O}\left(\frac{d}{\varepsilon}\right).$$

$\qquad\square$

Recall the condition Equation (7) for an $(\alpha, \gamma, p)$-well-conditioned basis.

**Lemma 3.7.** *Let $X \in \mathbb{R}^{n \times d}, y \in \mathbb{N}_0^n$ be a $\rho$-complex dataset, i.e., Equation (2) holds. Let $p \in \{1, 2\}$. Let $\lambda \geq 1$ be the slope parameter from either Lemma 2.2 or Lemma 2.3 depending on the value of $p$. Let $\gamma$ be a conditioning parameter and $\eta > 0$ be arbitrary. Then the sensitivity for each $x_i$ with $x_i \beta > \eta$ is bounded by $\varsigma_i \leq \lambda \rho \gamma^p \|U_i\|_p^p + 2/n$. Their total sensitivity is bounded by $\mathfrak{S} \leq O\left(\rho d \sqrt{y_{\max}/\log(y_{\max})} + \log\log(1/\eta)\right)$ for $p = 1$, and $\mathfrak{S} \leq O(\rho d + \log(y_{\max}) + \log\log(1/\eta))$ for $p = 2$.*

*Proof of Lemma 3.7.* **Case 1:** $y_i = 0$: We start with the special case where $y_i = 0$. Recall that $x_i \beta > 0$. Then,

$$g_{y_i}(x_i \beta) = (x_i \beta)^p \leq \|U_i\|_p^p \|R\beta\|_q^p \leq \|U_i\|_p^p (\gamma \|URβ\|_q)^p$$
$$= \gamma^p \|U_i\|_p^p \|URβ\|_p^p = \gamma^p \|U_i\|_p^p \|X\beta\|_p^p = \gamma^p \|U_i\|_p^p \sum_{j=1}^n g_{y_j}(x_j \beta).$$

Next, we consider $0 \neq y_i \in \mathbb{N}$, and divide this into two sub-cases.

**Case 2(i):** $y_i \in \mathbb{N}, \eta < x_i \beta \leq y_i^{1/p}$:

We start with the case $0 < \eta \leq x_i\beta \leq y_i^{1/p}$.

For $g_{y_i}$ defined in Equation (4),

$$g_{y_i}(x_i\beta) := (x_i\beta)^p - py_i\log(x_i\beta) + \log(y_i!)$$

it holds using Lemma 2.1, the bounds on $x_i\beta$ and the monotonicity of $g_y(z)$ in the interval that

$$1 \leq g_{y_i}(x_i\beta) \leq \eta^p + y_i\log\left(\frac{1}{\eta^p}\right) + \log(y_i!) =: UB(y_i).$$

Let $\mathcal{G} = \{i \in [n] \mid 1 \leq g_{y_i}(x_i\beta) \leq UB(y_{\max})\}$. We subdivide $\mathcal{G} = \dot{\bigcup}_{j=1}^l \mathcal{G}_j$ into disjoint sets

$$\mathcal{G}_j = \{i \in \mathcal{G} \mid UB(y_{\max}) \cdot 2^{-j} < g_{y_i}(x_i\beta) \leq UB(y_{\max}) \cdot 2^{-j+1}\}, \ j \in \{1, \ldots, l\}.$$

Since $g_{y_i}(x_i\beta) \in [1, UB(y_{\max})]$, there can be at most $l \leq \lceil\log_2(UB(y_{\max}))\rceil$ groups. So

$$l \leq \log_2\left(\eta^p + y_{\max}\log\left(\frac{1}{\eta^p}\right) + \log(y_{\max}!)\right) \leq O\left(\log(y_{\max}) + \log\log\left(\frac{y_{\max}}{\eta}\right)\right)$$

Now let $n_j = |\mathcal{G}_j|$. We can bound the sensitivity (see Definition A.2) for each summand $g_{y_i}(x_i\beta)$ for $i \in \mathcal{G}_j$ by

$$\varsigma_i := \sup_\beta \frac{g_{y_i}(x_i\beta)}{\sum_{i=1}^n g_{y_i}(x_i\beta)} \leq \sup_\beta \frac{g_{y_i}(x_i\beta)}{\sum_{i \in \mathcal{G}_j} g_{y_i}(x_i\beta)} \leq \frac{UB(y_{\max}) \cdot 2^{-j+1}}{n_j \, UB(y_{\max}) \cdot 2^{-j}} = \frac{2}{n_j}$$

Summing over $i \in \mathcal{G}$ yields

$$\sum_{i \in \mathcal{G}} \varsigma_i = \sum_{j=1}^l \sum_{i \in \mathcal{G}_j} \frac{2}{n_j} = \sum_{j=1}^l \frac{2n_j}{n_j} = 2l \leq O\left(\log(y_{\max}) + \log\log\left(\frac{y_{\max}}{\eta}\right)\right)$$
$$\leq O\left(\log(y_{\max}) + \log\log\left(\frac{1}{\eta}\right)\right).$$

**Case 2(ii):** $y_i \in \mathbb{N}$, $x_i\beta > y_i^{1/p}$:

Now we take care of the remaining region where $x_i\beta > y^{1/p}$.

In particular, by Lemmas 2.2 and 2.3 for some scaling $\lambda = \lambda_p$ that depends on $p \in \{1, 2\}$, we have that

$$\frac{(z - y_i^{1/p})^p}{\lambda} \leq g_{y_i}(z) \leq z^p$$

in that region.

Let $UR$ be a decomposition of $X$, so that $x_i\beta = U_iR\beta$, and $U$ is again a $p$-well conditioned basis, in the sense of Equation (7).

Now, using our assumption given in Equation (2), we have the following inequalities

$$g_{y_i}(x_i\beta) \leq (x_i\beta)^p \leq \|U_i\|_p^p \|R\beta\|_q^p \leq \gamma^p \|U_i\|_p^p \|UR\beta\|_p^p$$
$$= \gamma^p \|U_i\|_p^p \sum_{j=1}^n (x_j\beta)^p \leq \rho\gamma^p \|U_i\|_p^p \sum_{j=1}^n (x_j\beta - y_j^{1/p})^p$$
$$\leq \lambda\rho\gamma^p \|U_i\|_p^p \sum_{j=1}^n g_{y_j}(x_j\beta).$$

Summing over all sensitivities, we get that the total sensitivity is bounded by

$$\mathfrak{S} \leq \rho\lambda(\alpha\gamma)^p + O\left(\log(y_{\max}) + \log\log\left(\frac{y_{\max}}{\eta}\right)\right).$$

For the first summand, there exists for each $p \in [1, 2]$, a so-called Auerbach basis attaining $\alpha = d, \gamma = 1$, see [43, Lemma 2.22]. For the special case $p = 2$, we even have that any orthonormal basis satisfies $\alpha = \sqrt{d}, \gamma = 1$ since $\|U\|_F = \sqrt{d}$, and for any $z \in \mathbb{R}^d$: $\|Uz\|_2 = \sqrt{z^T U^T U z} = \sqrt{z^T z} = \|z\|_2$.

Thus, in both cases we have suitable bases with $(\alpha\gamma)^p = d$.

For $p = 1$ Lemma 2.2 yields $\lambda \leq O(\sqrt{y_{\max}/\log y_{\max}})$. With this, the overall bound simplifies to

$$\mathfrak{S} \leq O\left(\rho d\sqrt{y_{\max}/\log y_{\max}} + \log\log(1/\eta)\right).$$

For $p = 2$ we have that $\lambda = 1$ suffices by Lemma 2.3. With this, the overall bound simplifies to

$$\mathfrak{S} \leq O(\rho d + \log(y_{\max}) + \log\log(1/\eta)).$$

$\square$

**Theorem 3.8.** *Let $X \in \mathbb{R}^{n \times d}, y \in \mathbb{N}_0^n$ be a $\rho$-complex dataset, i.e., Equation (2) holds. We can compute a weighted coreset $(K, w) \in \mathbb{R}^{k \times d} \times \mathbb{R}_{\geq 0}^k$ for the pth-root-link Poisson regression problem with $p \in \{1, 2\}$ on $D(\eta) := \{\beta \in \mathbb{R}^d : \forall i, \ x_i\beta > \eta\}$. The size of the coreset is bounded by $k = \tilde{O}(\varepsilon^{-2}d \cdot \min\{d, \varepsilon^{-1}\log(n)\log(y_{\max})\} \cdot m)$, where*

$$m = \begin{cases} \rho d \log\log(d)\sqrt{y_{\max}/\log(y_{\max})} + \log\log(1/\eta) & p = 1 \\ \rho d + \log(y_{\max}) + \log\log(1/\eta) & p = 2. \end{cases}$$

*Proof of Theorem 3.8.* We put our bounds from Corollary 3.6 and Lemma 3.7 together into the main theorem of the sensitivity framework Proposition A.5. That is, we calculate the sensitivity upper bounds $s_i$, take a sample according to the distribution $p_i = s_i/\sum_{j=1}^n s_j$ of the respective size, and reweight them accordingly. Then, the calculated bounds on the VC dimension and total sensitivity yield a bound on the required size, such that Proposition A.5 yields with constant probability that the reweighted subsample gives a $(1 \pm \eta)$-approximation uniformly over $\beta \in D(\eta)$.

Since the Auerbach basis used in the sensitivity calculations of Lemma 3.7 can be expensive to compute depending on the value of $p$, we use more efficient approximation techniques here.

In the case $p = 2$, we use a sparse oblivious $\ell_2$ subspace embedding by [9], which was explicitly proven in [34, Lemma 2.14], to give a $(\sqrt{2d}, \sqrt{2}, 2)$-well-conditioned basis. This is within absolute constant factors to the $(\sqrt{d}, 1, 2)$-conditioning of the Auerbach basis. Thus, the complexities given in Lemma 3.7 do not change in $O$-notation.

In the case $p = 1$, we use a more recent technique introduced in [44], called $(\alpha, \gamma, p)$-well-conditioned spanning sets. This is a relaxation of the well-conditioned basis $U$ given in Equation (7), where $U \in \mathbb{R}^{n \times s}$ and $z \in \mathbb{R}^s$, are allowed to have slightly increased dimension $s > d$. We also note that we only need to bound norms of vectors of the form $X\beta$ in the columnspan of the data matrix whose rank is bounded by $d$. We thus require the bounds to hold only for $y \in \mathbb{R}^s, s > d$ that actually represent vectors $X\beta$ in a different basis. Other aspects of Equation (7) remain unchanged.

Our proof is nearly verbatim to [44, Theorem 1.11]. Their algorithm constructs a matrix $R \in \mathbb{R}^{d \times s}$. We set $U = XR \in \mathbb{R}^{n \times s}$. By [44, Lemma 4.1], this can be done with $s = O(d\log\log(d))$, such that each column $U^{(i)}$, for $i \in [s]$ satisfies $\|U^{(i)}\|_p = 1$, and for every $\|X\beta\|_p = 1$ there exists a vector $y \in \mathbb{R}^s$, such that $X\beta = Uy$ and $\|y\|_2 = O(1)$.

For $p = 1$, this yields that $U$ is an $(\alpha, \gamma, 1)$-well-conditioned spanning set, where $\alpha = O(d\log\log(d))$, and $\gamma = O(1)$. For the bound on $\alpha$, it holds that

$$\|U\|_1 = \sum_{i=1}^s \|U^{(i)}\|_1 = s = O(d\log\log(d)).$$

For the bound on $\gamma$, note that the Hölder dual for $p = 1$ is $q = \infty$. Now, it follows for every $y \in \mathbb{R}^s$ that represents any $X\beta$ as a linear combination of columns of $U$ that,

$$\|y\|_\infty \leq \|y\|_2 \leq O(1) = O(1)\|X\beta\|_1 = O(1)\|Uy\|_1.$$

As a consequence in the case $p = 1$, this computational result replaces the $d$ factor from the Auerbach basis in the proof of Lemma 3.7 by a factor $(\alpha\gamma)^p = O(d\log\log(d))$, as we have claimed. $\square$

# D  Proofs for main approximation result

Recall Equation (8):

$$\tilde{\beta}^* := \operatorname{argmin}_{\beta' \in D(\eta)} f(X\beta'), \quad \beta^* := \operatorname{argmin}_{\beta \in D(0)} f(X\beta).$$

**Lemma 4.1.** *It holds for sufficiently small $\eta > 0$ that*

$$f(X\beta^*) \leq f(X\tilde{\beta}^*) \leq (1 + O(\eta))f(X\beta^*).$$

*Proof of Lemma 4.1.* Recall from Section 2 that we choose $X$ to be the design matrix that includes an intercept, i.e., every row of $X$ is of the form $x = (1, x^{(1)}, x^{(2)}, \ldots, x^{(d-1)})$. Note that by definition we have that $D(\eta) \subset D(0)$ is a proper subset. Thus, $f(X\beta^*) \leq f(X\tilde{\beta}^*)$ follows immediately.

Next, define for every $\beta \in D(0)$ its shifted version to be in one-to-one correspondence with a unique $\beta' \in D(\eta)$ via the translation

$$\beta' := \beta + \eta e_1, \tag{15}$$

where $e_1$ is the first standard basis vector. Recall that we choose the first column of $X$ to be $(1, \ldots, 1) \in \mathbb{R}^n$.

Now consider $(\beta^*)' \in D(\eta)$ to be the shifted version of the global optimizer $\beta^* \in D(0)$. Since $\tilde{\beta}^*$ minimizes the loss function over $D(\eta)$, it follows that $f(X\tilde{\beta}^*) \leq f(X(\beta^*)')$.

Finally, we claim that there exists an absolute constant $C \geq 0$ such that for all sufficiently small $\eta > 0$ we have for all $\beta$ that Equation (9) holds, i.e.,

$$f(X\beta') \leq f(X\beta) + \eta^p n + \eta C f(X\beta).$$

By summing the result of Lemma 2.1 over all $n$ inputs, we have that $f(X\beta^*) \geq n$. Applying our claim to $\beta^*$ thus yields

$$f(X(\beta^*)') \leq f(X\beta^*) + \eta^p n + \eta C f(X\beta^*) \leq (1 + \eta + C\eta)f(X\beta^*) = (1 + O(\eta))f(X\beta^*).$$

This proves that Lemma 2.1 and Equation (9) imply the upper bound of the lemma.

It remains to prove our claim of Equation (9) for $p = 1$, and $p = 2$ separately.

**Case $p = 1$:** We have

$$
\begin{aligned}
f(X\beta') &= \sum_{i \in [n]} (x_i\beta + \eta) - y_i \log(x_i\beta + \eta) + \log(y_i!) \\
&\leq \sum_{i \in [n]} x_i\beta - y_i \log(x_i\beta) + \log(y_i!) + \eta = f(X\beta) + \eta n,
\end{aligned}
$$

which satisfies Equation (9) with $C = 0$.

**Case $p = 2$:** We have

$$
\begin{aligned}
f(X\beta') &= \sum_{i \in [n]} (x_i\beta)^2 + 2\eta x_i\beta + \eta^2 - 2y_i \log(x_i\beta + \eta) + 2y_i \log(x_i\beta) - 2y_i \log(x_i\beta) + \log y_i! \\
&= \sum_{i \in [n]} (x_i\beta)^2 + 2\eta x_i\beta + \eta^2 - 2y_i \log \frac{x_i\beta + \eta}{x_i\beta} - 2y_i \log(x_i\beta) + \log y_i! \\
&= f(X\beta) + \sum_{i \in [n]} 2\eta x_i\beta + \eta^2 - 2y_i \log \frac{x_i\beta + \eta}{x_i\beta} \\
&= f(X\beta) + \sum_{i \in [n]} 2\eta x_i\beta + \eta^2 - 2y_i \log \left(1 + \frac{\eta}{x_i\beta}\right) \\
&= f(X\beta) + n\eta^2 + 2\eta \sum_{i \in [n]} x_i\beta - \frac{y_i}{\eta} \log \left(1 + \frac{\eta}{x_i\beta}\right).
\end{aligned}
$$

Now using that $\log(1+x) \geq \frac{x}{1+x}, \forall x > -1$, we bound the error for every $y \in \mathbb{N}$ by a function $\phi = \phi_y$.

$$z - \frac{y}{\eta} \log\left(1 + \frac{\eta}{z}\right) \leq z - \frac{y}{\eta} \frac{\eta}{z} \frac{z}{z+\eta} = z - \frac{y}{z+\eta} =: \phi(z), \tag{16}$$

The first and second derivatives are given by

$$\phi'(z) = 1 + \frac{y}{(z+\delta)^2} \geq 1,$$

$$\phi''(z) = -\frac{2y(z+\delta)}{(z+\delta)^4} < 0,$$

from which we know that the function is monotonically increasing and concave. On the other hand, we know that $g_y(z)$ is convex, monotonically increasing on $z \in [y^{1/2}, \infty)$ and is bounded below by 1. We can thus show the claim for $C = 3$ by comparing the functions as well as their derivatives at $z = y^{1/2} + 1$.

First note that by monotonicity, we have for all $z \leq y^{1/2} + 1$ that

$$\phi(z) \leq \phi(y^{1/2} + 1) = \frac{(y^{1/2} + 1)^2 + \eta(y^{1/2} + 1) - y}{y^{1/2} + 1 + \eta} = \frac{y + 2y^{1/2} + 1 + \eta y^{1/2} + \eta - y}{y^{1/2} + 1 + \eta}$$

$$= \frac{y^{1/2} + 1 + \eta + (1+\eta)y^{1/2}}{y^{1/2} + 1 + \eta} = 1 + \frac{(1+\eta)y^{1/2}}{y^{1/2} + 1 + \eta}$$

$$\leq 1 + \frac{(1+\eta)y^{1/2}}{y^{1/2}} = 1 + 1 + \eta \leq 3 \cdot 1 \leq 3g(z).$$

In particular this holds for $z = y^{1/2} + 1$ as well.

It remains to compare the derivatives for the choice of $z = y^{1/2} + 1$. We have

$$\phi'(y^{1/2} + 1) = 1 + \frac{y}{(y^{1/2} + 1 + \delta)^2} \leq 1 + \frac{y}{(y^{1/2})^2} = 2,$$

which implies that we also have

$$3g'(y^{1/2} + 1) = 3 \cdot 2 \frac{(y^{1/2} + 1)^2 - y}{y^{1/2} + 1}$$

$$= 3 \cdot 2 \frac{y + 2y^{1/2} + 1 - y}{y^{1/2} + 1}$$

$$= 3 \cdot 2 \frac{2y^{1/2} + 1}{y^{1/2} + 1} \geq 3 \cdot 2 > 2 \geq \phi'(y^{1/2} + 1).$$

This completes the proof of $\phi(z) \leq 3g(z)$ for all $z$ by convexity of $g$ and concavity of $\phi$.

Overall, the claim of Equation (9) follows with $C = 3(p-1)$, for both $p \in \{1, 2\}$, which also concludes the proof of the lemma. $\qquad\square$

**Theorem 4.2.** *Let $\varepsilon \in (0, 1/14)$. Let $(C, w)$ be a coreset according to Theorem 3.8. Let $\tilde{\beta} :=$ $\text{argmin}_{\beta \in D(\varepsilon)} f_w(C\beta)$, $\beta^* := \text{argmin}_{\beta \in D(0)} f(X\beta)$. Then*

$$f(X\beta^*) \leq f(X\tilde{\beta}) \leq (1 + \varepsilon)f(X\beta^*).$$

*Proof of Theorem 4.2.* We invoke Lemma 4.1 with $\eta = \varepsilon$ to show that the shifted version of $\beta^*$, i.e., $\beta_{good} := (\beta^*)' = \beta^* + \eta e_1$ is a $(1 + \varepsilon)$-approximation and $\beta_{good} \in D(\eta)$. Thus, the optimizer $(\beta')^* \in D(\eta)$ cannot be worse than a $(1 + \varepsilon)$-approximation.

The coreset construction of Theorem 3.8 works uniformly over $D(\eta)$. It thus yields a coreset $C \subset X$ of size $k$ with weights $w \in \mathbb{R}^k$ such that if we denote the weighted loss on the coreset by $f_w(C\beta)$, it satisfies

$$\forall \beta \in D(\eta) : (1 - \varepsilon)f(X\beta) \leq f_w(C\beta) \leq (1 + \varepsilon)f(X\beta) \tag{17}$$

Then defining $\beta_{good} := \beta^* + \eta e_1$, we have $f(X\tilde{\beta}) \geq f(X\beta^*)$ since $D(\eta) \subset D(0)$. Moreover, assuming $0 < \varepsilon \leq \frac{1}{2}$ we have

$$f(X\tilde{\beta}) \leq \frac{1}{1-\varepsilon} f_w(C\tilde{\beta}) \leq \frac{1}{1-\varepsilon} f_w(C\beta_{good})$$

$$\leq \frac{1+\varepsilon}{1-\varepsilon} f(X\beta_{good}) \leq \frac{(1+\varepsilon)^2}{1-\varepsilon} f(X\beta^*) \leq (1+7\varepsilon)f(X\beta^*)$$

rescaling $\varepsilon$ finishes our main result. $\qquad\square$

# E   Proofs for lower bounds

**Lemma 6.1.** *Consider a number $n \geq 8$ of points equidistant on a unit circle in a 2-dimensional affine subspace embedded in $\mathbb{R}^d, d \geq 3$, each with label $y_i = 1$. Then the sensitivity of each point for the Poisson model with pth-root-link for $p \in \{1, 2\}$ is arbitrarily close to $1$. Consequently, any coreset for the Poisson regression model must comprise all $\Omega(n)$ input points.*

*Proof of Lemma 6.1.* We first note that our construction can be embedded arbitrarily in $d \geq 3$ dimensional spaces. For simplicity, we describe the construction for $d = 3$, where the first dimension corresponds to the affine translation and the other two describe the location in the 2-dimensional subspace.

Recall that our point set is given by $x_i = (1, \cos(\frac{2\pi i}{n}), \sin(\frac{2\pi i}{n}))$, $i \in [n]$, and $y_i = 1$ for every $i \in [n]$. By symmetry of the construction, it suffices to analyze w.l.o.g. the sensitivity of point $x_n = (1, \cos(2\pi), \sin(2\pi)) = (1, 1, 0)$. Since the sensitivity is defined as the supremum over all $\beta$, it also suffices to find one $\beta$ for which the sensitivity is arbitrarily close to $1$. To this end, for a small $\eta > 0$ yet to be determined, we choose

$$\beta = (1 + \eta, -\cos(2\pi), -\sin(2\pi))^T = (1 + \eta, -1, 0)^T,$$

where $1 + \eta$ represents a translation term and $(-1, 0)^T$ represents a 'normal' term that lives within the 2-dimensional subspace mentioned above. This normal term defines a hyperplane $H$ (which is in fact a line) within the 2-dimensional subspace. The normal points towards the center of the point set, and the hyperplane $H$ is at distance exactly $x_n\beta = 1 + \eta - 1 = \eta$ from $x_n$.

A simple trigonometric calculation yields that the separation between $x_n$ and the neighboring points $x_1$ and $x_{n-1}$ along the direction orthogonal to $H$ is exactly $1 - \cos(2\pi/n)$. Since $n \geq 8$, it holds that $(2\pi/n)^2/3 \leq 1 - \cos(2\pi/n) \leq (2\pi/n)^2/2$ by a second order Taylor series expansion of the cosine function. All other points are even farther away from $x_n$ than $x_1$ and $x_{n-1}$, and therefore also further from $H$. Also note that if we let $x'_n = (1, -1, 0)$ be the antipodal point of $x_n$ on the circle, we see that the distances of all points from $H$ are less than $x'_n\beta = 1 + \eta + 1 = 2 + \eta < 3$.

Recall that for arbitrary $p \geq 1$, the function $g_y$ defined in Equation (4) is minimized at $y^{1/p}$, which in this case equals $y_i = 1$. We have that roughly half of the points are at distance at least $1 + \eta$ and distance at most $3$ from $H$. By strict convexity, we have that $g_1$ is also strictly increasing on the interval $[1, \infty)$. We can thus upper bound the contribution of each of these points by at most

$$g_1(x_i\beta) \leq \frac{n}{2} g_1(3) = (3^p - p\log(3)) \leq 9 - 1 = 8 \leq 8\log(n). \tag{18}$$

For the other half of the points (except $x_n$) the contribution is upper bounded by the loss that occurs closest to $H$. By strict convexity again, we have that $g_1$ is also strictly decreasing on the interval $(0, 1]$. We argued that the points are sufficiently separated, so we get that each of their contributions is bounded by

$$g_1(x_i\beta) \leq g_1(1 - \cos(2\pi/n)) \leq g_1((2\pi/n)^2/3) \leq g_1(1/n^2)$$

$$\leq 1/n^{2p} - 2p\log(1/n) = 1/n^2 + 2p\log(n)$$

$$\leq 1 + 4\log(n) \leq 8\log(n). \tag{19}$$

Now, choosing $\eta = \exp(-n^2)$, we have that the cost of the point $x_n$ is lower bounded by

$$g_1(x_n\beta) = g_1(\eta) \geq \left(\frac{1}{\exp(n^2)}\right)^p + p\log(\exp(n^2)) \geq n^2.$$

Thus, we have that

$$\varsigma_n = \sup_{\beta'} \frac{g_1(x_n \beta')}{\sum_{i=1}^n g_1(x_i \beta')} \geq \frac{n^2}{n^2 + 8n \log(n)} \xrightarrow{n \to \infty} 1.$$

Since there is no sensitivity upper bound below 1 that holds for arbitrarily large $n$ and for each point, [41, Lemma A.1] implies that the coreset must comprise all $\Omega(n)$ points. $\qquad\square$

**Lemma 6.2.** *Let $\Sigma_D$ be a data structure for $D = [X, y] \in \mathbb{R}^{n \times d} \times \mathbb{R}^n$, $d \geq 3$, that approximates negative log-likelihood queries $\Sigma_D(\beta)$ for Poisson regression with the pth-root-link for $p \in \{1, 2\}$, such that for some $\varphi \geq 1$ it holds that*

$$\forall \beta \in \mathbb{R}^d : f(X\beta) \leq \Sigma_D(\beta) \leq \varphi \cdot f(X\beta).$$

*If $\varphi < \frac{n}{8\log(n)}$ then $\Sigma_D$ requires $\Omega(n)$ bits of memory.*

*Proof of Lemma 6.2.* We reduce from the indexing problem for which we know that it has one-way randomized communication complexity $\Omega(n)$ [25]. We construct a protocol as follows. Alice is given a vector $b \in \{0, 1\}^n$. She produces for every $i$ with $b_i = 1$ the points $x_i = (1, \cos(\frac{2\pi i}{n}), \sin(\frac{2\pi i}{n}))$ in canonical order. The corresponding counts are set to $y_i = 1$. She builds and sends $\Sigma_D$ to Bob, whose task is to guess the bit $b_j$. Let the size of $\Sigma_D$ in bit complexity be $s(n)$ bits, and note that $s(n)$ corresponds to the amount of bits that have been communicated. Bob chooses to query $\beta = (1 + \eta, -\cos(\frac{2\pi j}{n}), -\sin(\frac{2\pi j}{n}))$.

By symmetry of the construction, we can assume w.l.o.g. that the upper bounds Equations (18) and (19) on the costs in the proof of Lemma 6.1 continue to hold.

Thus, if $b_j = 0$, then $x_j$ does not exist and the cost of all other points is bounded from above by

$$f(X\beta) \leq 8n \log(n).$$

If $b_j = 1$, then $x_j$ is at distance exactly

$$x_j \beta = \left(1, \cos\left(\frac{2\pi j}{n}\right), \sin\left(\frac{2\pi j}{n}\right)\right) \cdot \left(1 + \eta, -\cos\left(\frac{2\pi j}{n}\right), -\sin\left(\frac{2\pi j}{n}\right)\right)^T$$

$$= 1 + \eta - \cos\left(\frac{2\pi j}{n}\right)^2 - \sin\left(\frac{2\pi j}{n}\right)^2 = 1 + \eta - 1 = \eta.$$

Thus, choosing $\eta = \exp(-n^2)$, the cost is bounded below by $f(X\beta) \geq g_1(\eta) \geq n^2$ as in the proof of Lemma 6.1.

Given that $\varphi < \frac{n^2}{8n \log(n)} = \frac{n}{8 \log(n)}$, Bob can distinguish these two cases based on the data structure only, by deciding whether $\Sigma_D(\beta)$ is strictly smaller or larger than $n^2$. Consequently, it holds that $s(n) \geq \Omega(n)$, since this solves the indexing problem. $\qquad\square$

**Lemma 6.3.** *For all $x \in [-1/e, 0)$, it holds that $W_0(x) \leq \sqrt{2(1 + ex)} - 1$.*

*Proof of Lemma 6.3.* First we claim that

$$\tau - \log(1 + \tau) \geq -\frac{1}{2} \log(1 - \tau^2) = -\log(\sqrt{1 - \tau^2}), \quad \tau \in (-1, 0]. \tag{20}$$

Define $l(\tau) := \tau - \log(1 + \tau) + \frac{1}{2} \log(1 - \tau^2)$. Then $l(0) = 0$. The derivative of $l$ is

$$l'(\tau) = 1 - \frac{1}{1 + \tau} - \frac{1}{2} \cdot \frac{2\tau}{1 - \tau^2} = \frac{1 - \tau^2 - 1 + \tau - \tau}{1 - \tau^2} = \frac{-\tau^2}{1 - \tau^2} < 0, \quad \forall \tau \in (-1, 0]$$

which implies that for every $\tau \in (-1, 0]$ we have $l(\tau) \geq 0$. This proves Equation (20). Next, we follow the proof strategy of [38, Theorem 3.2]: define a new variable $\tau = \tau(x) := -(W_0(x) + 1)$, so that $-W_0(x) = 1 + \tau$. Since $W_0(x)e^{W_0(x)} = x$ for $x \geq -1/e$, the definition of $\tau$ implies that $(1 + \tau)e^{-(1+\tau)} = -x$. Let $x \in [-1/e, 0)$ be arbitrary. Then since $W_0(0) = 0$ and $W_0(-1/e) = -1$, we have $\tau(x) \in (-1, 0]$, and thus

$$(1 + \tau)e^{-(1+\tau)} = -x \iff \quad \tau - \log(1 + \tau) = -\log(-x) - 1$$

$$\implies \quad -\log(\sqrt{1-\tau^2}) \le -\log(-x) - 1 \qquad \text{by Equation (20)}$$

$$\iff \quad 1 + \log(-x) \le \log(\sqrt{1-\tau^2}).$$

The last inequality is equivalent to

$$1 \le \log\left(\frac{\sqrt{1-\tau^2}}{-x}\right) \iff e \le \frac{\sqrt{1-\tau^2}}{-x} \iff -ex \le \sqrt{1-\tau^2}.$$

Now since $(\sqrt{1-\tau^2})^2 = 1 - \tau^2 \le 1 - \tau^2 + (\frac{\tau^2}{2})^2 = (1 - \frac{\tau^2}{2})^2$ for every $\tau \in (-1, 0]$, it follows that

$$-ex \le 1 - \frac{\tau^2}{2} \iff \frac{\tau^2}{2} \le 1 + ex \iff |W_0(x) + 1| \le \sqrt{2(1 + ex)},$$

where the rightmost equivalence above follows from the definition $\tau := -(W_0(x)+1)$. The inequality on the right-hand side above implies the desired conclusion. $\qquad\square$

Recall the function $g_{y_i}$ defined in Equation (4).

**Lemma 6.4.** *Let $y \in \mathbb{N}$ be arbitrary, $p = 1$, and $\tau = y^{1/p}$ in the definition Equation (4) of $g_y$. Let $h_\lambda(z) := \frac{(z-y^{1/p})^p}{\lambda}$ for $z > 0$. Then $g_y$ and $h_\lambda$ are tangent to each other if and only if $\lambda = \lambda^*(y) = (W_0(\frac{-y}{(y!)^{1/y}\exp(2)})+1)^{-1}$, in which case the unique tangent point is $z^*(y) = \frac{y\lambda^*(y)}{\lambda^*(y)-1}$. In addition, $\lambda^*(y) = \Theta(\sqrt{y_{\max}/\log(y_{\max})})$.*

*Proof of Lemma 6.4.* A point of tangency $\tilde{z}$ of the curves $g_y$ and $h_\lambda$ is defined as a point where the functions agree and their derivatives agree. To identify the point where the derivatives of $g_y$ and $h_\lambda$ agree, we observe

$$g_y'(\tilde{z}) = 1 - \frac{y}{z} = \frac{1}{\lambda} = h_\lambda'(\tilde{z}) \iff 1 - \frac{1}{\lambda} = \frac{y}{\tilde{z}} \iff \tilde{z} = \frac{\lambda y}{\lambda - 1}.$$

Since $g_y(z) \ge \frac{1}{2}\log(2\pi y)$ and $h_\lambda(z) < 0$ for $z < y$ if $\tau = y$, the tangent point cannot lie in the interval $(0, y]$. Hence, $\tilde{z} > y$ must hold. Combining this observation with the equation $\tilde{z} = \frac{\lambda y}{\lambda - 1}$ for the point where the derivatives agree, we conclude that $\lambda > 1$.

Now suppose that $\tilde{z}$ is a point where the functions $g_y$ and $h_\lambda$ agree:

$$h_\lambda(\tilde{z}) = g_y(\tilde{z})$$

$$\iff \quad y\log(\tilde{z}) - \tilde{z} + \frac{\tilde{z}}{\lambda} = \frac{y}{\lambda} + \log(y!)$$

$$\iff \quad y\log(\tilde{z}) - \tilde{z}\frac{\lambda - 1}{\lambda} = \frac{y}{\lambda} + \log(y!)$$

$$\iff \quad \log(\tilde{z}) - \tilde{z}\left(\frac{\lambda y}{\lambda - 1}\right)^{-1} = \frac{1}{\lambda} + \frac{1}{y}\log(y!)$$

$$\iff \quad \tilde{z}\exp\left(-\tilde{z}\left(\frac{\lambda y}{\lambda - 1}\right)^{-1}\right) = \exp\left(\frac{1}{\lambda}\right)(y!)^{1/y}$$

$$\iff \quad \left(-\tilde{z}\left(\frac{\lambda y}{\lambda - 1}\right)^{-1}\right)\exp\left(-\tilde{z}\left(\frac{\lambda y}{\lambda - 1}\right)^{-1}\right) = -\left(\frac{\lambda y}{\lambda - 1}\right)^{-1}\exp\left(\frac{1}{\lambda}\right)(y!)^{1/y}.$$

If $\tilde{z}$ is a point of tangency, then by the equivalent condition above for $g_y' = h_\lambda'$, it follows that $\tilde{z} = \frac{\lambda y}{\lambda - 1}$. Substituting this into the last equation above yields

$$-1\exp(-1) = -\left(\frac{\lambda y}{\lambda - 1}\right)^{-1}\exp\left(\frac{1}{\lambda}\right)(y!)^{1/y}$$

$$\iff \quad -\frac{y}{(y!)^{1/y}}\exp(-1) = -\frac{\lambda - 1}{\lambda}\exp\left(\frac{1}{\lambda}\right)$$

$$\Longleftrightarrow \qquad -\frac{y}{(y!)^{1/y}}\exp(-2)=\left(\frac{1}{\lambda}-1\right)\exp\left(\frac{1}{\lambda}-1\right).$$

The last equation above yields that

$$\frac{1}{\lambda}-1=W_k\left(\frac{-y}{(y!)^{1/y}\exp(2)}\right),\quad k\in\mathbb{Z}.\tag{21}$$

We can further specify the branches of the Lambert W function as follows. By definition of the factorial, $1\le\frac{y}{(y!)^{1/y}}$ for all $y\in\mathbb{N}$. On the other hand, by Stirling's approximation,

$$\frac{\exp(1-\frac{1}{12y^2})}{(2\pi y)^{1/(2y)}}<\frac{y}{(y!)^{1/y}}<\frac{\exp(1-\frac{1}{12y^2+y})}{(2\pi y)^{1/(2y)}}<\exp(1),\quad y\in\mathbb{N}\setminus\{1\}.\tag{22}$$

Hence,

$$-\exp(-1)<-\frac{y}{(y!)^{1/y}}\exp(-2)\le-\exp(-2),\quad y\in\mathbb{N}.$$

This implies that the argument of the Lambert W function in Equation (21) lies in the interval $(-e^{-1},-e^{-2}]$. By definition of the Lambert W function, this in turn implies that in Equation (21), we need to consider only $k=0$ and $k=-1$, which means that

$$\frac{1}{\lambda^*}=\frac{1}{\lambda^*(y)}\in\left\{W_k\left(\frac{-y}{(y!)^{1/y}\exp(2)}\right)+1\;:\;k\in\{0,-1\}\right\}$$

For all $x\in[-\exp(-1),-\exp(-2)]$, $W_0(x)\ge W_{-1}(x)$, with equality holding only for $x=-\exp(-1)$. This follows from the definition of $W_0$ as the principal branch of the Lambert W function. Hence,

$$\lambda^*(y)=\frac{1}{W_0\left(\frac{-y}{(y!)^{1/y}\exp(2)}\right)+1},\quad y\in\mathbb{N},$$

which proves the first statement of Lemma 6.4.

Next, we show that $\lambda^*(y)=\Theta(\sqrt{y_{\max}/\log(y_{\max})})$. We use a lower bound for the principal branch $W_0$ of the Lambert W function for negative arguments from [38, Theorem 3.2]:

$$(ex+1)^{1/2}-1\le W_0(x),\quad\forall x\in[-e^{-1},0].\tag{23}$$

Since we showed above that the argument of the Lambert W function in Equation (21) lies in the interval $(-e^{-1},-e^{-2}]$, we may apply Equation (23).

Combining Lemma 6.3 with Equation (23), implies that for all $x\in[-e^{-1},0]$,

$$\sqrt{ex+1}-1\le W_0(x)\le\sqrt{2(1+ex)}-1\Leftrightarrow\frac{1}{\sqrt{2(1+ex)}}\le\frac{1}{W_0(z)+1}\le\frac{1}{\sqrt{(1+ex)}}.$$

By Equation (22), we may thus set $x=-\frac{y}{(y!)^{1/y}}e^{-2}$ in the above inequalities, which yields

$$2^{-1/2}\left(\frac{-y}{(y!)^{1/y}}e^{-1}+1\right)^{-1/2}\le\frac{1}{W_0\left(\frac{-y}{(y!)^{1/y}}e^{-2}\right)+1}=\lambda^*(y)\le\left(\frac{-y}{(y!)^{1/y}}e^{-1}+1\right)^{-1/2}\tag{24}$$

for all $y\in\mathbb{N}$. Again by Equation (22),

$$1-\exp\left(-\frac{1}{12y^2+y}-\frac{1}{2y}\log(2\pi y)\right)<\frac{-y}{(y!)^{1/y}}e^{-1}+1<1-\exp\left(-\frac{1}{12y^2}-\frac{1}{2y}\log(2\pi y)\right).$$

Note that

$$0<\frac{1}{12y^2+y}+\frac{1}{2y}\log(2\pi y)<\frac{1}{12y^2}+\frac{1}{2y}\log(2\pi y).$$

Since $y\mapsto\frac{1}{12y^2}+\frac{1}{2y}\log(2\pi y)$ is decreasing on $[1,\infty)$ and has the value $\frac{1}{12}+\frac{\log(2\pi)}{2}<\frac{1}{2}$ at $y=1$, it suffices to consider the quantity $1-\exp(-x)$ for $x\in[0,\frac{1}{2}]$. By Taylor's approximation, we have

$$\frac{x}{2}<1-\exp(-x)<x,\quad\forall x\in[0,1]\tag{25}$$

and applying the lower bound in Equation (25) to the lower bound for $-\frac{y}{(y!)^{1/y}}e^{-1}+1$ below Equation (24) yields

$$\frac{-y}{(y!)^{1/y}}e^{-1}+1 > 1 - \exp\left(-\frac{1}{12y^2+y} - \frac{1}{2y}\log(2\pi y)\right)$$

$$> \frac{1}{2}\left(\frac{1}{12y^2+y} + \frac{1}{2y}\log(2\pi y)\right)$$

$$= \frac{1}{2}\cdot\frac{1+6(y+1/12)\log(2\pi y)}{12y^2+y}$$

for all $y \in \mathbb{N}$. Applying this to the upper bound for $\lambda^*(y)$ in Equation (24) yields for all $y \in \mathbb{N}$

$$\lambda^*(y) \leq \left(2\cdot\frac{12y^2+y}{1+6(y+1/12)\log(2\pi y)}\right)^{1/2} \leq \left(\frac{26y^2}{6y\log(2\pi y)}\right)^{1/2} = O\left(\sqrt{\frac{y}{\log(y)}}\right).$$

Applying the upper bound in Equation (25) to the upper bound for $-\frac{y}{(y!)^{1/y}}e^{-1}+1$ below Equation (24) yields

$$\frac{-y}{(y!)^{1/y}}e^{-1}+1 < \frac{1}{12y^2} + \frac{1}{2y}\log(2\pi y) = \frac{1+6y\log(2\pi y)}{12y^2} < \frac{12y\log(2\pi y)}{12y^2}$$

where the rightmost inequality follows since the function $y \mapsto 6y\log(2\pi y)$ is increasing on $[1, \infty)$ and has a value strictly larger than 1 at $y = 1$. Applying the inequality above to the lower bound for $\lambda^*(y)$ in Equation (24) yields

$$\lambda^*(y) \geq \left(\frac{1}{2}\frac{y^2}{y\log(2\pi y)}\right)^{1/2} = \Omega\left(\sqrt{\frac{y}{\log(y)}}\right), \quad \forall y \in \mathbb{N}.$$

This completes the proof that $\lambda^*(y) = \Theta(\sqrt{y_{\max}/\log(y_{\max})})$. $\qquad\square$

**Lemma 6.5.** *Let $p \in \mathbb{N}$, $p \geq 3$. Then there does not exist an absolute constant $C \geq 0$ such that for all sufficiently small $\eta > 0$ and for all $\beta \in D(0)$, $\beta' := \beta + \eta e_1 \in D(\eta)$ satisfies*

$$f(X\beta') \leq f(X\beta) + \eta^p n + \eta C f(X\beta). \tag{9}$$

*Proof of Lemma 6.5.* First note that

$$f(X\beta') = \sum_{i\in[n]}(x_i\beta+\eta)^p - py_i\log(x_i\beta+\eta) + \log(y_i!)$$

$$= \sum_{i\in[n]}\left((x_i\beta)^p + \eta^p + \sum_{\ell=1}^{p-1}\binom{p}{\ell}(x_i\beta)^\ell\eta^{p-\ell}\right) - py_i\log(x_i\beta+\eta) + \log(y_i!)$$

$$= f(X\beta) + \eta^p n + \sum_{i\in[n]}\sum_{\ell=1}^{p-1}\binom{p}{\ell}(x_i\beta)^\ell\eta^{p-\ell} - py_i\log\left(\frac{x_i\beta+\eta}{x_i\beta}\right),$$

where the last equation follows by the definition of $f(X\beta)$, given the hypothesis on $p$. Define for every $y \in \mathbb{N}$ the auxiliary function

$$\varphi_{1,y,p}(z) := \frac{1}{\eta}\left(\sum_{\ell=1}^{p-1}\binom{p}{\ell}z^\ell\eta^{p-\ell} - py\log\left(\frac{z+\eta}{z}\right)\right), \quad z > 0. \tag{26}$$

Then the inequality Equation (9) is equivalent to

$$\sum_{i\in[n]}\varphi_{1,y_i,p}(x_i\beta) \leq C\sum_{i\in[n]}g_{y_i}(x_i\beta) \iff 0 \leq \sum_{i\in[n]}Cg_{y_i}(x_i\beta) - \varphi_{1,y_i,p}(x_i\beta),$$

which implies the following statement: there exists $C, \eta^* > 0$ such that for every $n \in \mathbb{N}$, $(y_i)_{i\in[n]} \in \mathbb{N}^n$, $\eta \leq \eta^*$, and $(z_i)_{i\in[n]} \in \prod_{i\in[n]}[y_i^{1/p}, \infty)$, it holds that

$$0 \leq \sum_{i\in[n]}Cg_{y_i}(z_i) - \varphi_{1,y_i,p}(z_i). \tag{27}$$

This statement yields the following necessary condition for Equation (9): there exists $C, \eta^* > 0$ such that for every $n \in \mathbb{N}$, $(y_i)_{i \in [n]} \in \mathbb{N}^n$, $\eta \le \eta^*$, and $(z_i)_{i \in [n]} \in \prod_{i \in [n]} [y_i^{1/p}, \infty)$, at least one summand on the right-hand side must satisfy $0 \le C g_{y_i}(z_i) - \varphi_{1, y_i, p}(z_i)$. This is because if every summand in the sum in Equation (27) were strictly negative, then the sum itself must be strictly negative. If the necessary condition above does not hold, then by considering the contrapositive, we conclude that Equation (9) does not hold either.

We now show that the necessary condition does not hold, by proving that for every $C, \eta > 0$ and $n \in \mathbb{N}$, there exists $(y_i)_{i \in [n]} \in \mathbb{N}^n$ such that for every $i \in [n]$, $C g_{y_i}(y_i^{1/p}) - \phi_{1, y_i, p}(y_i^{1/p}) < 0$. Indeed, for every $i \in [n]$, suppose that every $y_i \in \mathbb{N}$ satisfies

$$\frac{2C}{p(p-1)\eta} < \frac{y^{(p-2)/p}}{\frac{1}{2}\log(2\pi y) + \frac{1}{12y}}. \tag{28}$$

By the hypothesis that $p \ge 3$ and by the fact that the denominator of the right-hand side grows more slowly than the numerator, it follows that there exist infinitely many values of $y_i$ that satisfy Equation (28). Thus it remains to show that Equation (28) implies $C g_y(y^{1/p}) - \phi_{1, y, p}(y^{1/p}) < 0$.

By the inequality $\frac{x}{1+x} \le \log(1 + x)$, $x > 0$, we obtain

$$-\frac{y}{z} = -\frac{y}{\eta}\frac{\eta}{z} \le -\frac{y}{\eta}\log\left(1 + \frac{\eta}{z}\right), \quad \forall \eta, z > 0.$$

Rewrite the auxiliary function $\varphi_{1, y, p}$ from Equation (26) and bound it from below, first by using the lower bound above, and then by using the hypothesis that $p \in \mathbb{N}$, $p \ge 3$:

$$\varphi_{1, y, p}(z) = \sum_{\ell=1}^{p-1} \binom{p}{\ell} z^\ell \eta^{p-\ell-1} - p\frac{y}{\eta}\log\left(\frac{z + \eta}{z}\right)$$

$$\ge \sum_{\ell=1}^{p-1} \binom{p}{\ell} z^\ell \eta^{p-\ell-1} - p\frac{y}{z} > pz^{p-1} + \frac{p(p-1)}{2}z^{p-2}\eta^1 - p\frac{y}{z}.$$

Setting $z = y^{1/p}$, it thus suffices to show that

$$C g_y(y^{1/p}) < p\left(y^{(p-1)/p} + \eta\frac{p-1}{2}y^{(p-2)/p} - y^{1-1/p}\right) = p\eta\frac{p-1}{2}y^{(p-2)/p}. \tag{29}$$

Using (4) to evaluate $g_y(y^{1/p})$ and using the upper bound on $\log(y!)$ from Stirling's approximation, we conclude that $g_y(y^{1/p}) < \frac{1}{2}\log(2\pi y) + \frac{1}{12y}$. Now Equation (28) implies Equation (29), because

$$C\left(\frac{1}{2}\log(2\pi y) + \frac{1}{12y}\right) < p\eta\frac{p-1}{2}y^{(p-2)/p} \iff \frac{2C}{p(p-1)\eta} < \frac{y^{(p-2)/p}}{\frac{1}{2}\log(2\pi y) + \frac{1}{12y}}.$$

This completes the proof. $\qquad\square$

## F   On the shifted domain and feasibility

In this section, we show that the restriction to $D(\eta)$ in Theorem 3.8 and Section 4 do not lead to feasibility issues.

First recall that we had defined for any $\eta \ge 0$

$$D(\eta) := \{\beta \in \mathbb{R}^d \ : \ \forall i, \ x_i\beta > \eta\}.$$

Also recall that for the ID- and square root-link — and in fact for any $p$th-root-link — the loss function as defined in Equations (3) and (4) includes a $\log(x\beta)$ term, which restricts the feasible region to $\beta$ such that for all $x_i, i \in [n]$ it holds that $x_i\beta > 0$. Thus $\beta \in D(0)$ is the natural domain induced by the model. In particular, this restriction is not our choice.

Our domain shift idea restricts the domain even further to $\beta \in D(\eta) \subset D(0)$, for $\eta > 0$. Clearly, some solutions that are feasible in the problem formulated over $D(0)$ are no longer feasible in the

problem formulated over $D(\eta)$. But as we prove in Appendix D, we can construct a coreset that holds for all $\beta \in D(\eta)$, and $D(\eta)$ contains at least one $\beta$ that is a $(1 + \varepsilon)$-approximation for the optimal solution $\beta^*$ of the problem on the original domain $D(0)$, and evaluated on the full dataset. These two parts are combined to prove that the final minimizer $\tilde{\beta} \in D(\eta)$ optimized on the coreset is a $(1 + O(\varepsilon))$-approximation compared to the value of $\beta^*$, when both are evaluated on the full dataset.

In the other direction, no infeasible solution can become feasible because of the proper subset relation $D(\eta) \subset D(0)$. Finally, note that for any data and any fixed $\eta < \infty$, both, $D(\eta)$ and $D(0)$ are non-empty, since they consist of all $\beta$ that parameterize hyperplanes that put the convex hull of input points (respectively, the additive $\eta$-inflation of the convex hull of input points) in the positive open halfspace. Thus there always exist feasible solutions, which means that no instance can become completely infeasible by means of our methods.

# G    Pseudocode, data and experimental results

## G.1    Pseudocode

Here we give pseudocode for our coreset construction Algorithm 1 and for the subsequent optimization procedure Algorithm 2:

---

**Algorithm 1** Coreset algorithm for $p$th-root-link Poisson regression.

---

**Input:** data $X \in \mathbb{R}^{n \times d}, Y \in \mathbb{N}_0^n$, number of rows $k$ (see Theorem 3.8).
**Output:** coreset $C = (X', Y', w) \in \mathbb{R}^{k' \times d} \times \mathbb{N}^{k'} \times \mathbb{R}^{k'}$ with $k' = k + |\text{Ext}(X)|$ rows
1: Let $X_{CH}$ be the extreme points $\text{Ext}(X)$ on the convex hull of $X$ or their $\varepsilon$-kernel approximation (cf. Section 5), let $Y_{CH}$ be their corresponding labels
2: Assign weight vector $w_{CH} = 1$ corresponding to all points in $X_{CH}, Y_{CH}$
3: Let $X = X \setminus X_{CH}, Y = Y \setminus Y_{CH}$
4: **if** $p = 1$ **then**
5:     Calculate a well-conditioned spanning set $B$ (see [44, Lemma 4.1])
6:     Set $Q = XB$
7: **else** for $p = 2$
8:     Sketch the data to obtain $\tilde{X} = \Pi X$ (see [9])
9:     Calculate the $QR$ decomposition of the sketch $\tilde{X} = \tilde{Q}R$
10:     Set $Q = XR^{-1}$ (see [15, 34])
11: Approximate the $\ell_p$ sensitivities by $s_i := \|Q_i\|_p^p + 1/n$
12: Sample $k$ rows of $X$ and $Y$ i.i.d. with probability $p_i = s_i / \sum_j s_j$ to obtain $X_{core} \in \mathbb{R}^{k \times d}$ and their corresponding labels $Y_{core} \in \mathbb{N}^k$
13: Set $w_{core} \in \mathbb{R}^k$ such that $w_{core,j} = 1/(kp_i)$ if sample $j$ corresponds to row $i$
14: Concatenate $X_{CH}$ with $X_{core}$ to obtain $X'$
15: Concatenate $Y_{CH}$ with $Y_{core}$ to obtain $Y'$
16: Concatenate $w_{CH}$ with $w_{core}$ to obtain $w$
17: **return** $C = (X', Y', w)$

---

---

**Algorithm 2** Domain shift optimizer for $p$th-root-link Poisson regression.

---

**Input:** data $X \in \mathbb{R}^{n \times d}, Y \in \mathbb{N}_0^n$, error parameter $\varepsilon \in (0, \frac{1}{3})$.
**Output:** $(1 + \varepsilon)$-approximate solution $\tilde{\beta}$.
1: Run Algorithm 1 with input $X, Y$ and $k$ as specified to obtain the coreset $(X', Y', w)$. Let $X_{CH}$ be defined as in Algorithm 1.
2: Run any convex optimization algorithm to find the optimal solution $\tilde{\beta}$ for the $p$th-root-link Poisson regression objective on $(X', Y', w)$ under the constraint that $\forall x_i \in X_{CH}: x_i \beta > \varepsilon$ (see Theorem 4.2)
3: **return** $\tilde{\beta}$.

---

## G.2 Synthetic data generation

We generated for each $p \in \{1, 2\}$ a dataset with dimensions $n = 100\,000, d = 7$ with $n$ labels corresponding to each point.

- Construction of $X$:
  1. Start with 6 standard basis vectors $(z_i)_{i=1}^6$ and add the all zero vector $z_0 = 0$ to be the extreme points on the convex hull.
  2. Construct a matrix in $\mathbb{R}^{(n-7)\times d}$ with i.i.d. standard Gaussian entries. Translate and rescale the rows of this matrix so that the resulting rows lie in the interior of the convex hull of the points $(z_i)_{i=0}^6$. Concatenate the matrix with the resulting to the matrix with rows given by $(z_i)_{i=0}^6$ generated by the first step. Call the resulting matrix $Z \in \mathbb{R}^{n \times d}$.
  3. Horizontally concatenate a column of length $n$ consisting only of ones to the left of the matrix $Z$ (add an intercept) to get $X \in \mathbb{R}^{n \times (d+1)}$.

- Construction of $\beta$:
  1. Draw one sample $\widetilde{\beta} \sim 10^{1/p} \cdot N(0, I_d)$, with $d = 6$ as above.
  2. Find $\min_{i \in [n]}(Z\widetilde{\beta})_i$ for $Z$ from the construction of $X$
  3. Compute $b := \max\{1, 2^{1/p} \cdot |\min_i(Z\widetilde{\beta})_i|\}$
  4. Define $\beta := (b, \widetilde{\beta}) \in \mathbb{R}^{d+1}$.

- Construction of $y$:
  1. Compute $\lambda := (X\beta)^p \in \mathbb{R}^n$
  2. For each $i = 1, \ldots, n$, draw $Y_i \sim \mathrm{Poisson}(\lambda_i)$, and store the resulting vector as $y$.

## G.3 Experimental illustration

All experiments were run on a commodity machine with Intel Core i7-7700K processor (4 cores, 4.2GHz, 32GB RAM) and took overall around 50 minutes to complete. The Python code of [34] was adapted to the Poisson regression setting.[4] We applied it with the appropriate $p \in \{1, 2\}$ to the datasets with dimensions $n = 100\,000, d = 7$ generated as detailed in the previous section.

We compared our method to uniform sampling as a baseline, which is widely popular due to its simplicity and general applicability.

We varied the reduced size between 50 and 600 in equal increments of size 50. For each reduced size and each method, we performed 201 independent repetitions.

Our results are shown in Figure 1. The red (Poisson with $p$th-root-link) and blue (uniform sampling) solid lines display the median approximation ratio across 201 independent repetitions for each reduced size. The shaded areas below and above the solid lines indicate $2\times$ standard errors of the respective medians.

For both $p \in \{1, 2\}$ the results look widely similar, although the case $p = 1$ is slightly more distinctive. We thus focus our further description on the case $p = 1$.

Our novel Poisson subsampling method generally outperforms uniform sampling, and their $2\times$ standard error intervals are very narrow.

The shaded blue area *under* the blue solid line indicates that the $2\times$ standard error for uniform sampling is slightly more narrow than the $2\times$ standard error for 1-Poisson regression, i.e., Poisson regression with ID-link. However, this error is optimistically calculated only on repetitions that succeed in providing a valid approximation when applied to the original full data.

The shaded blue area *above* the blue solid line is unbounded, which indicates that some of the repetitions yield solutions that are infeasible for the original full data problem, and thus fail to give an appropriate approximation. Even in a few feasible cases, approximation ratios were $1.5 - 2.5 \times 10^9$, which may be explained by missing points that are very close to the boundary of the convex hull, thus causing huge errors. The fraction of repetitions leading to infeasible solutions was always

---

[4]Our new code is available at `https://github.com/Tim907/poisson-regression/`.

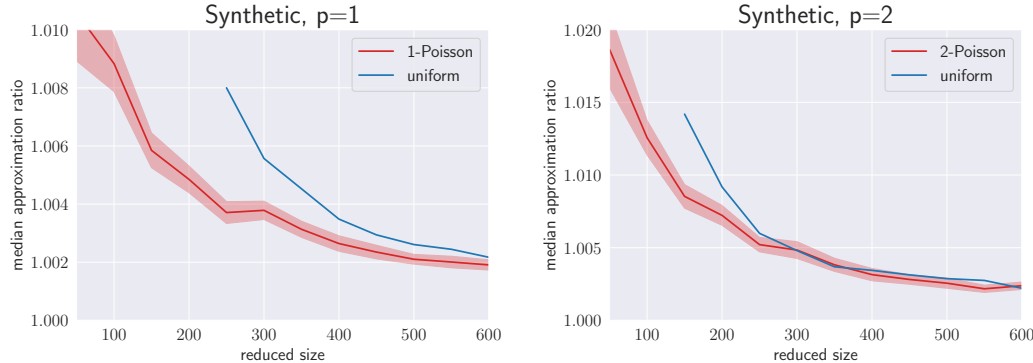

Figure 1: Experimental results for two synthetic data sets with $p = 1$ (left), respectively $p = 2$ (right). Our method is presented in red and compared against uniform sampling, which is presented in blue. Solid lines indicate the median and shaded areas indicate $\pm 2$ standard errors around the median taken across 201 independent repetitions for each reduced size between 50 and 600 in equal increment steps of 50. For the blue shaded area below the blue solid line, only feasible repetitions were counted, while the blue shaded area above represents the unbounded standard error without this restriction. For some lower reduced sizes, even the median was infinite, which results in an interrupted blue solid line. This indicates that more than half of the repetitions gave infeasible results when using uniform sampling with low sample sizes, while our method never produced infeasible results.

non-negligible and we note that the solid blue line was interrupted below a reduced size of $250$ (respectively $150$ for $p = 2$) meaning that even the median was infinite, indicating that more than half of the repetitions of uniform sampling were infeasible for the original problem.

In contrast to that, our method produced feasible results in all repetitions, across all reduced sizes, confirming our discussion in Appendix F, and giving approximation ratios very close to 1.

