# OpenReview forum: "Data subsampling for Poisson regression with pth-root-link"
_NeurIPS.cc/2024/Conference — NeurIPS 2024 poster_

### Official Review · Reviewer_L5wz · 2024-06-12

**Soundness:** 3
**Presentation:** 2
**Contribution:** 3
**Rating:** 6
**Confidence:** 1

**Summary:**

the paper considers sampling for Poisson regression, making explicit the dependence of the size of coreset on various parameters, in particular the effect of different link functions. The paper has a theoretical flavor

**Strengths:**

the results seems to be new

**Weaknesses:**

Frankly I am not familiar with this particular approach. One possible weakness may be that it is not clear how sharp this result is. It is also not clear what characteristics of Poisson regression make the study entirely different from many other regression models. I submit this particular review early so the chair may decide whether to look for additional reviewers.

**Questions:**

none

**Limitations:**

seems there is little discussion on limitation

---

> ### Author Rebuttal · Authors · 2024-08-05
>
> In the last line of our abstract (lines 18-20) we state that we show the limitations of our analysis for $p$-th degree root link functions for $p\geq 3$, and state that these limitations show the need for other methods if one aims to generalize our approach to this range of values of $p$. We repeat this point in lines 62-65. We also indicated on the NeurIPS Paper Checklist that Section 6 contains lower bounds that indicate the limitations of constructing coresets for $p$-th power link Poisson models in general, and the limitations of our specific approach and analysis, for instance the limitations regarding input/complexity parameters such as $y_{\max}$. We also point to [23] for limitations regarding the canonical log link.
>
> Our lower bounds on the parameters, together with linear VC dimension and linear sensitivity, and linear $\rho$ dependence leave no room for improvement if one uses the sensitivity framework. We also refer to the related work Section 1.3, where previous finite sample size results were either unbounded or $O(\sqrt{d})$ error, instead of our $1+\varepsilon$. We will add some comments on this.

---

> > ### Comment · Reviewer_L5wz · 2024-08-09
> >
> > I have read the response and have no comment

---

### Official Review · Reviewer_HZmN · 2024-07-01

**Soundness:** 3
**Presentation:** 2
**Contribution:** 3
**Rating:** 7
**Confidence:** 4

**Summary:**

Coresets are a technique in efficient algorithms for data analysis in which a dataset is compressed into a weighted subset of its examples. Typically, coresets are constructed such that a loss function for a given optimization problem (e.g. linear regression, logistic regression, or Poisson regression for this work) is preserved up to a $(1\pm\epsilon)$ factor error. This work extends this line of work to handle the Poisson regression case.

While the most popular choice of the log-link does not admit small coresets, the authors study the case of the ID and square root link functions and show that for these log-link functions, Poisson regression admits small coresets, assuming the smallness of a complexity parameter $\rho$ which they introduce building on prior work on logistic regression.

**Strengths:**

This work shows that under certain natural (and necessary) modifications, Poisson regression admits small coresets. Coresets are a popular topic of study in the recent machine learning literature, and this work makes progress on understanding coresets for Poisson regression.

**Weaknesses:**

The first main weakness of this work is that there are many modifications and assumptions that must be made to the Poisson regression in order to obtain small coresets. As the authors mention, the canonical log-link for Poisson regression does not admit small coresets, and this is a standard fact in the literature (e.g. formalized in [23], as the authors cite). Thus, the authors instead consider a polynomial link function, but the authors do not provide sufficient arguments as to why this is still an interesting problem to study, either from a theory or practice perspective. Indeed, once the growth rate of the function is polynomial and one assumes a balancedness condition, then small coresets can be constructed using known techniques in theory [24]. While this type of result may still be interesting if the regression problem is widely studied, Poisson regression with noncanonical links seems too artificial of a setting to be interesting for a wide audience.

The second main weakness is that the techniques largely are a straightforward generalization of techniques from [23, 24, 25] and the conclusions do not seem surprising given the prior work.

**Questions:**

How are Lemmas 6.1 and 6.2 different from the lower bound argument of Theorem 6 in [23]?

**Limitations:**

See weaknesses.

---

> ### Author Rebuttal · Authors · 2024-08-05
>
> We do not agree with the statement that Poisson regression with links other than the canonical log link is 'artificial'. The identity link was applied in epidemiology, see e.g. [a], or [b]. The root link function has been applied to forecasting for queueing systems [c], and to account for misspecification bias in maximum likelihood estimation [d]. When the estimated mean count of the data is zero, then the canonical log link causes problems that can be avoided by using the root link; see e.g. the paragraph "Use of the square root (sqrt) link function" in Section 5.4 of [e]. There are also informal discussions on stackexchange that shed more light on the choice of link function, which we unfortunately are not allowed to link in our response.
>
> While the general outline follows the same structure as previous papers (in fact almost all papers based on the sensitivity framework share this outline), the details are quite different. First note that the change of link function changes the norms to relate to from $\ell_\infty$-norm to $\ell_1/\ell_2$. The reviewer seems to suggest that this choice of polynomial functions resolves the problem given previous literature, which is indeed not the case as our linear lower bounds prove. Indeed, the fact that even after chosing polynomial functions, the loss still depends on the convex hull (i.e. $\infty$-norm) is somewhat challenging in our analysis.
> Next, the 'balancedness' $\mu$ assumption in [24,25] is needed due to the asymmetry of logistic/probit losses. However, in our paper, the asymmetry of Poisson loss is taken care of by our *novel domain shift* idea to avoid large contributions. It is not taken care of by the $\rho$ parameter!
> Also, $\rho$ has a different interpretation that is natural for Poisson, but not applicable to logit nor probit. It balances the mean and the variance, and motivates 'moving' the polynomial lower envelopes. Without this, we stress that the slope $\lambda$ parameter would need to be $y_{\max}/\log(y_{\max})$ since with growing $y$ the loss function 'moves' polynomially along the $x\beta$ axis but only logarithmically along the $g_y(x\beta)$ axis. See our high level discussion in lines 83-89.
> Indeed, though not obvious from the current writeup, this induces a $\sum_{i \in[n]} y_i =\Omega(n)$ dependence. So it is a completely different difficulty that the supposedly 'not surprising' or 'straightforward' analysis addresses by the balancing condition. The most common point between $\rho$ and $\mu$ is the fact that both correspond to or arise naturally from their respective statistical modeling.
> Please also consider the further discussion and the separation between the $p\in\\{1,2\\}$ cases with respect to the $y_{\max}$ dependence. These are further points that were previously completely unexplored and led to novel and interesting non-trivial results regarding the important Lambert $W_0$ function as a byproduct.
> So while some basic parts were borrowed from previous literature that we cited, we kindly ask the reviewer to recognize as well our non-trivial extensions and modifications that go beyond the polynomial growth and $\ell_p$ space approximation of [24,25] and other previous literature.
>
> Lemmas 6.1 and 6.2 are not significantly different from the lower bound argument of Theorem 6 in [23]. Indeed, we had already stated in lines 96-97 that these results are adapted from previous literature, admittedly without specifying Lemmas 6.1 and 6.2 exactly. This was an oversight on our part and we will clarify the connection to Theorem 6 in [23]. While the construction of the bad dataset is similar to [23] and is widely standard across previous literature as in e.g. [25,27], [f], [g], **each** of these references as well as ours adapt the construction to the peculiarities of their loss function under study. In particular, [23] puts the hyperplane into the convex hull to detect the query point being 'far' on the expensive exponential side of the loss (other points are on the cheaper linear side). In our case, we put the hyperplane outside the convex hull, to detect the query point being 'close' to the hyperplane, so that its negative logarithmic loss dominates the polynomial loss of all other points. While this can in hindsight be considered a minor modification, with the same reasoning, we should reject a lot of previous NeurIPS, ICML, AAAI publications for building on the IJCAI paper [g] (and maybe even older references).
>
> [a] I. C. Marschner (2010). "Stable Computation of Maximum Likelihood Estimates in Identity Link Poisson Regression", Journal of Computational and Graphical Statistics, 19(3), 666–683. DOI: 10.1198/jcgs.2010.09127
>
> [b] Donna Spiegelman, Ellen Hertzmark (2005). "Easy SAS Calculations for Risk or Prevalence Ratios and Differences", American Journal of Epidemiology, Volume 162, Issue 3, Pages 199–200. DOI: 10.1093/aje/kwi188
>
> [c] Haipeng Shen. Jianhua Z. Huang (2008). "Forecasting time series of inhomogeneous Poisson processes with application to call center workforce management", Ann. Appl. Stat. 2 (2) 601 - 623. DOI: 10.1214/08-AOAS164
>
> [d] B. Efron (1992). "Poisson Overdispersion Estimates Based on the Method of Asymmetric Maximum Likelihood", Journal of the American Statistical Association, 87(417), 98–107. DOI: 10.1080/01621459.1992.10475180
>
> [e] J. Maindonald, W. J. Braun (2010). "Data analysis and graphics using R: an example-based approach" (4th ed.). Cambridge University Press. DOI: 10.1017/CBO9781139194648
>
> [f] J. Huggins, T. Campbell, T. Broderick (2016). "Coresets for scalable Bayesian logistic regression". Advances in neural information processing systems, 29. NeurIPS 2016.
>
> [g] Sariel Har-Peled, Dan Roth, Dav Zimak (2007). "Maximum Margin Coresets for Active and Noise Tolerant Learning". 836-841, Proceedings of the 20th International Joint Conference on Artificial Intelligence. IJCAI 2007.

---

> > ### Comment · Reviewer_HZmN · 2024-08-08
> >
> > Thank you for the rebuttal. The additional references on the fact that the identity and root link functions are actually used in practice are extremely valuable to me, please consider including it in the draft. I have raised my score based on that.
> >
> > I still don't quite see your point about the balancedness parameter $\rho$ not being the driving force behind the coreset bound. In the logistic regression setting, having a small $\mu$ parameter allows the coreset to be small, while the circle lower bound instance that is always used does not have a small $\mu$ parameter and thus has a $\Omega(n)$ lower bound on the coreset size [25]. Are you saying that the circle lower bound gives an $\Omega(n)$ lower bound despite a small $\rho$ parameter?
> >
> > The domain shift idea seems to a technique of avoiding a pathological $\epsilon$-neighborhood of 0 in the loss function to facilitate the analysis, which still contains a $(1+\epsilon)$-approximate minimizer. Indeed, this has not been needed in prior works on sensitivity sampling for GLMs since the loss functions have no asymptotes in prior work. I did not realize this on my first read. Some follow-up questions on feasibility issues: When using the canonical log link, we never have to worry about anything like this since the log link ensures that the expectation of the GLM will be positive. How is this handled for the identity and square root link? And can't the domain restriction turn a feasible instance into an infeasible one? If so, can small coresets still exist and this work's solution avoids them, or are small coresets just not possible for such instances? This should be considered and discussed more carefully. **These are very interesting questions, and I will consider increasing my score to a weak accept depending on how the authors answer these questions. Note that I don't expect all of these questions to be fully answered, as long as a thoughtful discussion is provided.**
> >
> > Tangential point: I found the discussion of the $\lambda$ parameter to be confusing/not well-explained. I now understand it to be a parameter which parameterizes a lower bound on the loss function.
> >
> > Re: Lemmas 6.1 and 6.2, I now realize I missed that these lower bounds were for the p-th root versions rather than for the log link version as in [23]. I agree that these lower bounds need to be slightly modified for each coreset problem being studied. The question was not to suggest that including this lower bound is a weakness (although it wouldn't be a stand-alone contribution either), I just wanted to clarify that it couldn't have just been cited from [23].

---

> > > ### Author Response · Authors · 2024-08-09
> > >
> > > Thank you again for your thoughtful comments and questions, and for appreciating our work more than before. Of course we will add the additional references to motivate the model.
> > >
> > > Regarding $\rho$ and $\lambda$: The short answer is yes, the circle lower bound gives $\Omega(n)$ even in the case where $\rho$ is small.
> > >
> > > We try to give a more detailed explanation of the role of $\rho$ (and $\lambda$):
> > > Recall in logistic regression [25] the (individual) loss function satisfies $z \leq g(z) \leq 2z$ for sufficiently large $z$. I.e. the upper and lower bounds are balanced up to a factor $2$ independent of $\mu$ (but $\mu$ is used to introduce balance between large (positive) and small (negative) contributions to avoid the circle lower bound).
> > >
> > > Now let us take a look at Poisson with ID link (arguments are similar for $p=2$). We would like to bound $z/\lambda \leq g_y(z) \leq z$ for sufficiently large $z$ for some value of $\lambda$. Now $\lambda$ would need to be roughly $y$ (ignoring logarithmic factors) because with growing $y$, the loss function widens, and its minimum moves mainly to the 'right', so we would need a very flat lower bound, so large $\lambda\approx y$. This would unfortunately give $\sum_{i \in[n]} y_i =\Omega(n)$ dependence in coreset size (not reflected in the circle lower bound).
> > >
> > > So instead, we additionally shift the lower bound: $(z-y)/\lambda \leq g_y(z) \leq z$ for sufficiently large $z$. This allows sublinear $\lambda$ and thus avoids $\Omega(n)$ (restricted to the large $z$). At the same time we need to relate the lower $(z-y)$ and upper bound $z$, which is the role of $\rho$. This shift and balancing assumption is not artificial or just used to make the calculations go through, but it is naturally consistent with the statistical model (see discussion below Equation (2)).
> > >
> > >
> > > Please note that the steps we described in the previous two paragraphs **only** help us to achieve the analogue in the Poisson regression setting of the bounds $z \leq g(z) \leq 2z$ that hold in the logistic regression setting, which is **just** for relating to the $\ell_1$-norm. This is consistent with the fact that the Poisson regression setting is more difficult than the logistic regression setting and indicates that the additional steps we have taken in our paper are nontrivial.
> > >
> > > All the above does **not** handle the asymptote near zero, which is the main source of hardness leveraged in the circle $\Omega(n)$ lower bound. The latter difficulty is later tackled by the domain shift idea.
> > >
> > > [To be continued in an additional comment.]

---

> > > ### Author Response · Authors · 2024-08-09
> > >
> > > Regarding feasibility: for the ID and square root link (in fact for any $p$th-root-link), recall that the loss function includes a $\log(x\beta)$ term, which restricts the feasible region to $\beta$ such that for all $x_i, i\in[n]$ it holds that $x_i \beta > 0$. So $\beta\in D(0)$ is the natural domain, induced by the model. This restriction is not our choice.
> > >
> > > The domain shift idea restricts the domain even further to $\beta\in D(\eta)\subset D(0)$, for $\eta > 0$. Clearly, some solutions that are feasible in the problem formulated over $D(0)$ are no longer feasible in the problem formulated over $D(\eta)$. But as we prove, we can construct a coreset that holds for all $\beta \in D(\eta)$, and $D(\eta)$ contains at least one $\beta$ that is a $(1+\varepsilon)$-approximation for the optimal solution $\beta^*$ of the problem on the original domain $D(0)$, and evaluated on the full dataset. These two parts are combined to prove that the final minimizer $\tilde\beta\in D(\eta)$ evaluated on the coreset is a $(1+O(\varepsilon))$-approximation for $\beta^*$.
> > >
> > > In the other direction, no infeasible solution can become feasible because of the proper subset relation $D(\eta)\subset D(0)$.
> > >
> > > Note that for any data and any fixed $\eta<\infty$, both $D(\eta)$ and $D(0)$ are non-empty, since they consist of all $\beta$ that parameterize hyperplanes that put the convex hull of input points (respectively, the additive $\eta$-inflation of the convex hull of input points) in the positive open halfspace. So there always exist feasible solutions, which means that no instance can become completely infeasible by our methods.
> > >
> > > Regarding the question of whether "small coresets [can] still exist ... just not possible for such instances?": We think that small coresets can `still exist' only under assumptions that make the problem of finding a coreset trivial. This is our reasoning: if an instance consists of the extreme points on the convex hull, and all but a small (sublinear) number of points are separated by an $\varepsilon$ distance to the boundary, then indeed the domain shift would not be necessary since the resulting domain structure is already in the data. But if the non-extremal points are allowed to get arbitrarily close to the boundary, and if we do **not** shift the domain, then we will not avoid high sensitivity points that are strictly inside the convex hull. Then the coreset size would necessarily depend on the distance of the non-extremal points to the boundary, and crucially on the number of points that are very close to the boundary of the convex hull, which again can be constructed to be $\Omega(n)$.

---

> > > > ### Comment · Reviewer_HZmN · 2024-08-09
> > > >
> > > > Thank you for the additional clarifications, these are very helpful, and the use of the intercept indeed makes everything make sense. I will raise my score to accept.

---

> ### Comment · Reviewer_HZmN · 2024-08-08
>
> Another comment: perhaps the authors could consider reflecting the fact that only $p$-th root links are considered in the title, since this paper does not give a solution to the canonical Poisson regression problem.
>
> Possibly dumb question: In the proof of Lemma 4.1, why is $\beta'\in D(\eta)$? it seems like if $X\beta \geq 0$ (pointwise), then it does not follow that $X\beta' = X\beta + \eta Xe_1\geq \eta$ depending on the value of $X$ on the first column.

---

> > ### Author Response · Authors · 2024-08-09
> >
> > We agree to changing the title to: "Data subsampling for Poisson regression with $p$th-root-link".
> > Please let us know if you have other suggestions in mind.
> >
> > We have already chosen $X$ to be the design matrix that includes an intercept. This means that the first column of $X$ consists of only ones. We stated this explicitly in line 119: $x=(1,x^{(1)},x^{(2)},\ldots,x^{(d-1)})$ (notation slightly changed in response to other reviewers). We will add a reminder of that fact in the proof of Lemma 4.1, where the reviewer pointed us.

---

### Official Review · Reviewer_73Rm · 2024-07-09

**Soundness:** 2
**Presentation:** 2
**Contribution:** 2
**Rating:** 6
**Confidence:** 3

**Summary:**

This paper demonstrates the theoretical bound analysis on data subsampling for Poisson regression with ID-link and root-link functions based on coreset method and the leverage of $\ell_r$ norms to the loss function. The $\text { ‘ } \rho \text {-complex’ }$ is a novel meaningful parameter for data compressibility under the context of subsampling and is a good starting point of theoretical construction.

**Strengths:**

This paper provides a way of subdividing the set of input functions into groups to obtain an improved $O(d)$ bound for VC-dimension. The overall bounds by combining VC dimension and total sensitivity is convincing and the proof is detailed for cases where $p \in\{(1,2)\}$.

**Weaknesses:**

This paper lacks a well-constructed structure, as it does not clearly introduce important concepts, provide a detailed discussion of related work, or include a summary or conclusion. To improve clarity, the proposed data subsampling techniques for Poisson regression might be presented in a pseudo-code format. While numerical simulation is not necessary, it would be a beneficial addition.

**Questions:**

The notation is a bit confusing for the Poisson regression models: is the discussion restricted to $p \in\{(1,2)\}$ cases so the notation in e.g., 119 line, are in scalar format? The notation should be more generalized as in Theorem 3.8.

**Limitations:**

The authors should consider restructuring the entire paper to clearly state the introduction, the proposed method along with its bound analysis, and the limitations and conclusion. Although the problem statement, proof, and results are meaningful, they lack clarity for the readers. Additionally, a major concern is the limited cases to which the analysis can be applied, specifically only for $p \in\{(1,2)\}$. This limitation significantly undermines the generalizability of the proposed method.

---

> ### Author Rebuttal · Authors · 2024-08-05
>
> 1. We have difficulty understanding what exactly is lacking in the structure of the paper. We remind the reviewer of the structure of our paper: In Section 1, we provide an introduction that introduces the crucial concepts of a $(1+\varepsilon)$-coreset and sublinear bounds on the coreset size. In Section 1.1, we list our contributions. In Section 1.2, we describe the high level ideas of our techniques --- for almost one page --- and introduce the important ideas that underpin our results. In Section 1.3, we give a representative sample of related work. In Section 2, we introduce more important concepts, e.g. the complexity parameter in (2) and the loss functions in (3) and (4) that are essential for the rest of the paper. In Section 3, we again introduce important concepts relating to VC dimension and the loss function from Section 2. After more than six pages of structured introduction of concepts, our first novel result appears on page 7 in Theorem 3.8, which gives an upper bound on the coreset size. In Section 4, we state our main approximation result, and introduce a novel shifting idea to overcome a difficulty posed by our loss function. There still remain difficulties for the subsequent optimization. We address these difficulties in Section 5, using ideas based on previous literature. In Section 6, we give further context to the upper bound on the coreset size from Theorem 3.8. We do this by providing lower bounds on the coreset size.
>
> The above shows that we introduced important concepts and presented our main results in a logical progression. Note also that we already summarized the contents of our paper in the abstract. We do not agree that a conclusion is absolutely necessary for a "well-constructed structure", but we would be willing to include a conclusion on a tenth page. For our initial submission, we had already moved as much content as possible to the supplementary and kept only the important results of our work in the main paper. Note that no other reviewer had issues with the structure of our paper.
>
> 2. The goal of a literature review in an 8-page paper is not to be detailed or exhaustive. Instead, the goal is to point the reader to representative examples of relevant literature, and to explain the differences between these examples and our present work, and this is exactly what we have done on lines 109-117. If the reviewer can point to specific examples of literature that they feel should have been included in the review, then we invite the reviewer to state these examples and to indicate why these suggestions are not adequately represented by the references we cite in lines 109-117.
>
> 3. We are willing to provide pseudocode in the supplementary material.
>
> 4. We agree that numerical results that provide proof of concept are not necessary but could be beneficial. We now have results of this kind and are willing to report them in the supplementary material.
> Please refer to our response to reviewer SQxM for more details regarding numerical results.
>
> We suspect that the confusion is due to an issue with our notation: on line 119, "$x=(1,x_1,\ldots,x_{d-1})\in\mathbb{R}^{d}$" indicates a row vector, whereas in Theorem 3.8, $x_i$ on line 277 indicates a row vector. We will change the notation of coordinates of $x\in\mathbb{R}^d$ to $x=(1,x^{(1)},x^{(2)},\ldots,x^{(d-1)})$ on line 119 and consistently write $x_i$ to denote the $i$-th row vector in the data matrix $X\in\mathbb{R}^{n\times d}$ as in Theorem 3.8.
>
> The reviewer describes the problem statement, proof, and results  as lacking clarity, and also as being meaningful. It is difficult for us to understand how both these statements can be true, since a necessary condition for a statement to be meaningful is that it is clear. We invite the reviewer to give one or two specific examples of results from our paper and to explain exactly what they do not understand about those results. Without this information, the reviewer's feedback appears vague and generic.
>
> We point out that the other reviewers who are familiar with the area of coresets had no issues with the clarity of our problem statements, proofs, or results.
>
> Our paper is about constructing coresets of suitable size for Poisson regression, and we achieve this goal for some link functions that satisfy two criteria: 1) they can be treated using a combination of known coreset construction methods and our novel techniques and results; and 2) they have been and continue to be used in statistics (see response to reviewer HZmN). At no point do we claim generalizability of the proposed method to all link functions or all $p$. For this reason, the reviewer's comment appears to be finding issue with aspects of the paper that are not relevant to our stated goals.
> On the contrary, our lower bound results (together with [23]) indicate the restriction to $p\in\\{1,2\\}$, and exclude $p\geq 3$, in particular the limiting case $p=\infty$ that corresponds to the canonical log-link. According to the reviewing guidelines for NeurIPS: "authors should be rewarded rather than punished for being up front about the limitations of their work" (see lines 18-20, and 62-65 for early mentions of the limitations). Moreover $p\in\\{1,2\\}$ are important standard choices in statistics (see response to reviewer HZmN).

---

> > ### Comment · Reviewer_73Rm · 2024-08-12
> >
> > I have read the authors' response and have no further comments.

---

### Official Review · Reviewer_SQxM · 2024-07-12

**Soundness:** 3
**Presentation:** 3
**Contribution:** 3
**Rating:** 7
**Confidence:** 3

**Summary:**

For Poisson regression, where the outcome variable is a positive integer, the paper provides sublinear coresets under certain assumption on the data characterized by parameter $\rho$. The link function is the $p^{th}$ root link function with $p = 1 , p=2$.  Without any assumptions the authors show that no sublinear size coresets are possible, in fact, they show lower bound results for any other data reduction techniques also. For $p = 1 , p=2$ and the parameter $\rho$, the coresets are constructed using the standard sensitivity framework which involves calculating sensitivity upper bounds and bounding the VC-dimension.  For other values of $p$, the authors show that their technique and analysis is not sufficient to get an $\epsilon$-coreset.

**Strengths:**

1) Though the paper is very similar in structure and high-level ideas to the paper "On Coresets for Logistic Regression", extending those ideas to the setting of Poisson regression is non-trivial. The paper appears solid in terms of theoretical contributions.

2) The writing of the paper is generally very good. Though the proofs are in the appendices, high level ideas and intuitions are provided in the paper for most of the results in the main part. The trick to shift the domain to get sensitivity upper bounds is neat and may find interest in the community working on coresets.

**Weaknesses:**

The only main weakness I think is the lack of any experimental results in the paper. Some small set of proof-of-concept experiments would have further strengthened the paper.

There are some small issues with writing. $x$ in $R^d$ is described in terms of its coordinates $1, x_1, x_2 \dots x_{d-1}$. However later $x_j$ is used to denote the $j^{th}$ vector in the dataset.

The writing in the final part of the paper appears rushed, may be due to space constraints. The authors talk about Lambert functions however there is no introduction as to what they are. It is difficult for readers not familiar with the concept.

**Questions:**

This may be basic question, maybe I am mission something, the feasible $\beta$ is such that $x_i\beta > 0$ is for the pth-root link or it will also hold in case of ID-link Poisson regression?

**Limitations:**

see weaknesses

---

> ### Author Rebuttal · Authors · 2024-08-05
>
> We will describe some proof-of-concept results that have been obtained since we submitted the paper for review.
> We have generated 6-dimensional data that consists of the vertices of a regular simplex (as extreme points on the convex hull) and $n$ further points from a normal distribution rescaled to be in the convex hull. Our resulting approximation factors are close to $1$ even for only $20-50$ subsamples except for a few repetitions where it fails, and gives factors $2-7$. In contrast, even for $500$ samples, uniform sampling fails even to produce feasible $\beta$ in one third of the cases, and even when results are feasible and the extreme points are included, some approximation ratios are in the order of $1.5-2.5 \times 10^9$ which are clearly explained by missing points that are very close to the boundary of the convex hull, thus causing huge errors. We will elaborate more on this and add corresponding numerical proof-of-concept in the supplementary material.
>
> We agree that the notation can be improved and will change the notation of coordinates of $x\in\mathbb{R}^d$ to $x=(1,x^{(1)},x^{(2)},\ldots,x^{(d-1)})$. We will continue to write $x_j$ to denote the $j$-th vector in the dataset.
>
> If by 'final part of the paper' the reviewer was referring to Section 6, then it is true that we had to remove some text from this section due to space constraints. In particular, we will give the definition of the Lambert function in the text and give some intuition as to why it is necessary to consider this function. Some intuition is already included in the last paragraph of Section 1.2.
>
> For ID-link Poisson regression, the set of feasible $\beta$ also consists of $\beta$ such that $x_i\beta >0$. This is because the loss function involves $\log (x_i\beta)$ for the ID link as well.

---

> > ### Comment · Reviewer_SQxM · 2024-08-09
> >
> > I have read the rebuttal.  I will keep my score

---

### Author Rebuttal · Authors · 2024-08-05

We would like to thank the reviewers for their time and valuable comments on our submission, which we would like to address in individual responses below.

---

### Decision · Program_Chairs · 2024-09-25

**Decision:**

Accept (poster)

**Comment:**

This paper studies the problem of constructing coresets for Poisson regression, obtaining sample complexity bounds that depend on a novel and natural complexity parameter of the problem. It builds on a growing line of work on constructing corsets for various “generalized linear models”, including, e.g., recent work on logistic regression. The paper has some clear draw backs, including a lack of experiments, limited new theoretical techniques, and it studies a non-canonical form of Poisson regression. Nevertheless, given the recent interest in coreset problems (e.g., due to their connection to agnostic active learning) we believe the paper will be of interest to many in the NeurIPS community, and all of the reviewers felt it should be accepted to the conference.